# Structural basis for xenobiotic extrusion by eukaryotic MATE transporter

Hirotake Miyauchi[1], Satomi Moriyama[2], Tsukasa Kusakizako[1], Kaoru Kumazaki[1], Takanori Nakane [1],
Keitaro Yamashita [3], Kunio Hirata[3], Naoshi Dohmae[4], Tomohiro Nishizawa[1], Koichi Ito[5], Takaaki Miyaji [6],
Yoshinori Moriyama[2], Ryuichiro Ishitani[1] & Osamu Nureki[1]

Mulitidrug and toxic compound extrusion (MATE) family transporters export xenobiotics to maintain cellular homeostasis. The human MATE transporters mediate the excretion of xenobiotics and cationic clinical drugs, whereas some plant MATE transporters are responsible for aluminum tolerance and secondary metabolite transport. Here we report the crystal structure of the eukaryotic MATE transporter from *Arabidopsis thaliana*, at 2.6 Å resolution. The structure reveals that its carboxy-terminal lobe (C-lobe) contains an extensive hydrogen-bonding network with well-conserved acidic residues, and their importance is demonstrated by the structure-based mutational analysis. The structural and functional analyses suggest that the transport mechanism involves the structural change of transmembrane helix 7, induced by the formation of a hydrogen-bonding network upon the protonation of the conserved acidic residue in the C-lobe. Our findings provide insights into the transport mechanism of eukaryotic MATE transporters, which is important for the improvement of the pharmacokinetics of the clinical drugs.

[1] Department of Biological Sciences, Graduate School of Science, The University of Tokyo, 2-11-16 Yayoi, Bunkyo-ku, Tokyo 113-0032, Japan. [2] Department of Membrane Biochemistry, Okayama University Graduate School of Medicine, Dentistry and Pharmaceutical Sciences, Okayama 700-8530, Japan. [3] RIKEN SPring-8 Center, Sayo-gun, Hyogo 679-5148, Japan. [4] Biomolecular Characterization Unit, RIKEN Center for Sustainable Resource Science, Wako, Saitama 351-0198, Japan. [5] Department of Computational Biology and Medical Sciences, Graduate School of Frontier Sciences, The University of Tokyo, Chiba 277-8562, Japan. [6] Advanced Science Research Center, Okayama University, Okayama 700-8530, Japan. Correspondence and requests for materials should be addressed to R.I. (email: ishitani@bs.s.u-tokyo.ac.jp) or to O.N. (email: nureki@bs.s.u-tokyo.ac.jp)

The extrusion of exogenous compounds is crucial for maintaining cellular homeostasis. Multidrug and toxic compound extrusion (MATE) family transporters are conserved in all domains of life, and export cationic compounds using the electrochemical gradient of $H^+$ or $Na^+$ across the plasma membrane. In bacteria, MATE transporters function primarily as xenobiotic efflux pumps, which confer multidrug resistance against various antibiotics in pathogens[1,2]. The human MATE transporters are ubiquitously expressed throughout the body, and are highly expressed in the brush-border membrane of the kidney and the bile canaliculi of the liver[3]. In these cells, MATE transporters export cationic drugs with diverse chemical structures, such as cimetidine, metformin, procainamide, cephalexin, and acyclovir[4,5], into urine and bile using the $H^+$ gradient[6], thereby determining the plasma concentrations of these drugs[3]. Moreover, fluoroquinolone antibiotics were demonstrated to be potent inhibitors of human MATE transporters, thus explaining the drug–drug interactions after the coadministration of cationic drugs and these antibiotics[4]. The platinum anticancer agent, cisplatin, is not a MATE transporter substrate, and thus its renal accumulation causes drug-induced nephrotoxicity[7], whereas oxaliplatin is a good substrate for the human MATE transporter, thereby explaining its lower renal toxicity. Thus, understanding the transport mechanism of human MATE transporters is important for improving the efficacies of these cationic drugs. Remarkably large numbers of MATE paralogues are encoded in plant genomes, and more than 50 MATE paralogues were found in *Arabidopsis thaliana*[8], suggesting their physiological importance. The plant MATE transporters mediate the export of diverse secondary metabolites, such as nicotine[9], flavonoids[10] and proanthocyanidin precursors[11], as well as xenobiotics[12]. Moreover, a MATE transporter is involved in the efflux of the plant growth hormone, abscisic acid, from vascular and guard cells in *A. thaliana*[13]. A subgroup of MATE transporters mediates citrate efflux from root cells, thereby conferring tolerance towards phytotoxic aluminum in acidic soils, which comprise 30% of Earth's ice-free land and thus constrain agricultural production[14–16].

MATE family transporters belong to the multidrug/oligosaccharidyl-lipid/polysaccharide (MOP) transporter superfamily[8], and are classified into the NorM, DinF and eukaryotic MATE (hereafter referred to as eMATE) subfamilies, based on amino-acid sequence similarity[17,18]. All of the prokaryotic MATE transporters are classified into the NorM and DinF subfamilies, whereas the eukaryotic transporters belong to the eMATE subfamily. Several crystal structures of the NorM and DinF subfamily transporters have been reported, and revealed that they consist of 12 transmembrane (TM) helices with N-lobe and C-lobe related by intramolecular pseudo twofold symmetry[19–23]. These structures provided insights into the transport mechanism of bacterial MATE transporters. The crystal structures of the NorM subfamily transporters demonstrated that NorM contains conserved acidic residues in the N-lobe and C-lobe, which are both important for the transport mechanism[19,21,24]. In contrast, the crystal structures of the DinF subfamily transporters showed that the conserved acidic residues clustered in the N-lobe are important for the substrate transport activity[20,22]. Furthermore, the crystal structures of *Pyrococcus furiosus* MATE (PfMATE), an archaeal DinF subfamily transporter, revealed that the bending of TM1 is coupled to the protonation of the N-lobe Asp residues, which is important for the substrate extrusion process[20].

An amino-acid sequence analysis of MATE family transporters demonstrated that eMATE and NorM share the conserved acidic residues in the C-lobe. In contrast, no conserved acidic residue is present in the N-lobe of eMATE, although acidic residues are commonly observed in the N-lobes of NorM and DinF. These observations suggested that the transport mechanism of eMATE shares some similarity to that of NorM, but is quite different from that of DinF[23]. However, the structural basis for the substrate recognition and the transport mechanism still remains elusive, as no structural information for the eMATE subfamily has been reported.

To elucidate the structural basis for the substrate transport mechanism by the eMATE subfamily, we determine the crystal structure of *A. thaliana* DTX14 (AtDTX14) at 2.6 Å resolution. Structure-based functional analyses of AtDTX14 and hMATE1 suggest the transport mechanism of the eMATE subfamily, in which the structural change of TM7 has an important role in the cationic substrate transport.

## Results

**Structure determination and overall structure of AtDTX14.** To investigate the transport mechanism of eMATE, we performed fluorescence-based screening[25] of the eMATE proteins from various eukaryotes. The results indicated that DTX14 from *A. thaliana* (AtDTX14), which shares ~ 32% sequence identity with human MATE1 (hMATE1) (Supplementary Fig. 1), would be suitable for the structural analysis (Supplementary Fig. 1). To verify the transport activity of AtDTX14, we performed the complementation analysis using the *E. coli* knock-out strain of drug exporter genes (6-KO strain). In this system, we measured the growth complementation of the 6-KO strain, in the presence of toxic compounds, by expressing the AtDTX14 gene (see "Methods"). The results showed that the expression of the AtDTX14 gene rescued the growth of the 6-KO strain in the presence of an antibiotic, norfloxacin (Fig. 1), indicating the norfloxacin export activity of AtDTX14. This result is consistent with a previous result that DTX1 from *A. thaliana*, which shares ~50% amino-acid sequence similarity with AtDTX14 (Supplementary Fig. 1), also has norfloxacin export activity[12]. Although the purified AtDTX14 protein exhibited excellent solution behavior, we were unable to crystallize the wild-type AtDTX14 protein. A previous structural analysis of PfMATE revealed that the mutation of the conserved Pro residue in TM1 (Pro26) to Ala improves the resolution of the crystals[20]. This Pro26 residue in TM1 is also conserved in AtDTX14 (Pro36), and thus we prepared the P36A mutant of AtDTX14. This P36A mutant exhibited decreased transport activity in the complementation analysis (Fig. 1), as also observed for the P26A mutant of PfMATE. The purified P36A mutant protein also exhibited excellent solution behavior, and we obtained crystals of the AtDTX14 P36A mutant that diffracted up to 2.6 Å resolution. The final structure was determined at 2.6 Å resolution, using the molecular replacement method (Table 1; Supplementary Fig. 2).

The overall structure of AtDTX14 is composed of 12 TM helices, which are arranged in a bilobed architecture consisting of the N-lobe (TM1-6) and the C-lobe (TM7-12) (Fig. 2a), as in the reported crystal structures of the bacterial MATE transporters. The N-lobe and C-lobe are connected by a long loop located on the cytosolic side. The N-lobe and C-lobe are related by intramolecular twofold pseudosymmetry, and are superimposable with an RMSD of 2.33 Å over 179 Cα atoms (Supplementary Fig. 3). The internal cavity is formed between the N-lobe and C-lobe, and is accessible from the extracellular space, but closed to the intracellular space mainly by the interactions between TM2 and TM8. Overall, AtDTX14 assumes a V-shaped structure open toward the extracellular side, similar to the other bacterial MATE structures. In contrast, the N-lobe and C-lobe cavities observed in the crystal structures of the previous bacterial MATE transporters are not present in the AtDTX14 structure (Fig. 2b). Moreover, the extracellular entrance of AtDTX14 is narrower than those of all

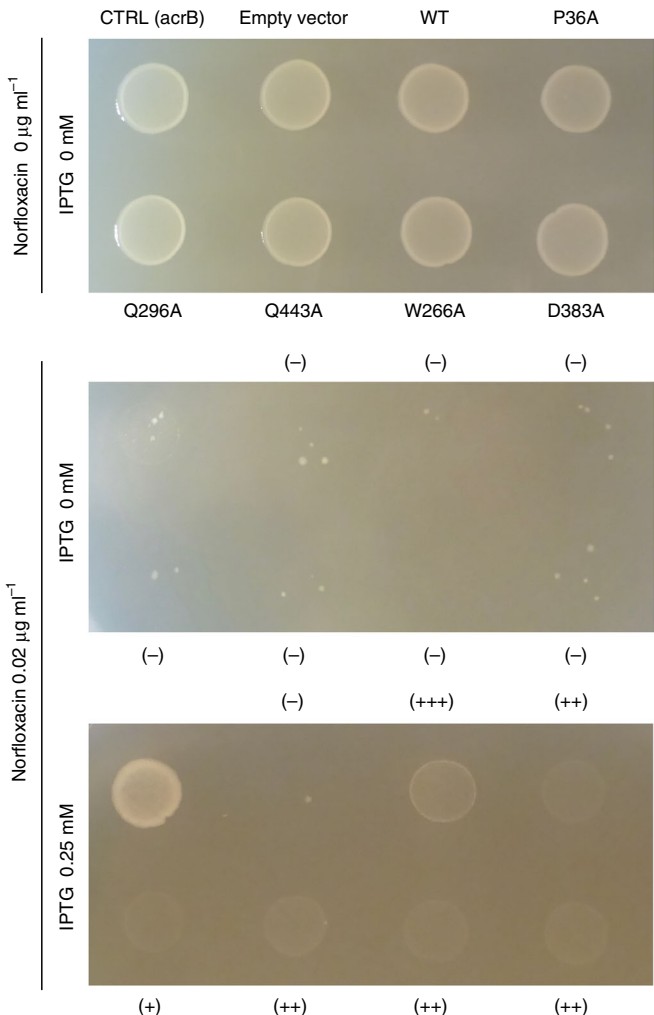

**Fig. 1** In vivo complementation analysis of AtDTX14. Norfloxacin-resistance growth assay of AtDTX14 and its mutants was performed, using the 6-knock-out E. coli strain. AtDTX14 expression was induced by 0 mM (top panel, non induction control, 12 h) and 0.25 mM IPTG, with 0 μg ml$^{-1}$ (middle panel, 18 h) or 0.02 μg ml$^{-1}$ (bottom panel, 12 h) of norfloxacin. The mutants that decreased the complementary activity are labeled with (+) or (++), and those that showed the wild-type AtDTX14 level of complementary activity are labeled with (+++). The transformant of the single knock-out strain (BW25113 acrAB::Δ macAB::Δ yojHI::Δ) harboring the isogenic wild-type acrB inducible expression vector was included as an induction control (CTRL)

**Table 1 Data collection and refinement statistics**

|  | AtDTX14 (PDB ID: 5Y50) |
| --- | --- |
| *Data collection* | |
| Beamline | SPring-8 BL32XU |
| Wavelength (Å) | 1 |
| No. of crystals | 100 |
| Space group | $P2_12_12_1$ |
| Cell dimensions | |
| $a$, $b$, $c$ (Å) | 52.8, 86.8, 116.4 |
| $\alpha$, $\beta$, $\gamma$ (°) | 90, 90, 90 |
| Resolution (Å) | 48.34–2.60 (2.76–2.60)$^a$ |
| $R_{meas}$ | 0.587 (6.833) |
| $I/\sigma(I)$ | 7.79 (0.90) |
| Completeness (%) | 98.5 (100) |
| Redundancy | 28.5 (28.5) |
| $CC_{1/2}$ | 0.994 (0.609) |
| | |
| *Refinement* | |
| Resolution (Å) | 50–2.6 |
| No. reflections | 16,814 |
| $R_{work}/R_{free}$ | 0.224/0.276 |
| No. of atoms | 3362 |
| Protein | 3362 |
| B-factors (Å$^2$) | |
| Protein | 54.4 |
| R.m.s. deviations | |
| Bond length (Å) | 0.003 |
| Bond angle (°) | 0.577 |
| Ramachandran plot | |
| Favored (%) | 98 |
| Allowed (%) | 2 |
| Outliers (%) | 0 |

$^a$Values in parentheses are for highest-resolution shell

bacterial structures, thus representing a partially occluded outward-open structure (Supplementary Fig. 4).

**Intracellular and extracellular gates**. On the intracellular side of the internal cavity, many interactions are formed between the N-lobe and C-lobe, thus isolating the internal cavity from the cytosol (Fig. 3a–c). A salt-bridge network is formed between Glu89 (TM2), Arg168 (TM4), and Arg394 (TM10) (Fig. 3b). Especially, the salt bridge between Glu89 in the N-lobe and Arg394 in the C-lobe may facilitate the closure of the intracellular gate. Furthermore, the intracellular side of the intracellular gate is closed by the interactions between TM2 and TM8. These interactions include the hydrogen bond between Asn96 of TM2 and Asn316 of TM8, as well as the hydrophobic interactions between Gly97 of TM2 and Gly319 of TM8 (Fig. 3c). Among these interactions in

the intracellular gate, the interactions between the Gly residues in TM2 and TM8 are also conserved in the bacterial MATE structures, such as NorM-VC[19] and PfMATE[20]. Moreover, these Gly residues are strictly conserved in the amino-acid sequences of the MATE family members (Supplementary Fig. 1). For all of the MATE family members, the side-by-side packing of the TM2 and TM8 helices facilitated by these conserved Gly residues may be important to tightly close the intracellular gate.

Previous reports proposed that the MATE family transporters operate by the rocker-switch-like mechanism, via the inward-open and outward-open conformations[19–23,26,27]. The recently reported structure of the lipid flippase, MurJ, supported this notion, exemplifying the inward-open conformation structure of the MOP superfamily[27]. Thus, the MATE transporters are also expected to form the inward-open structure, in which the internal cavity is open toward the cytosol and the extracellular gate is closed. To investigate the inward-open conformation of the eukaryotic MATE, we generated the inward-open model structure of AtDTX14, using the MurJ structure (Fig. 3d–f). No large steric clashes were observed in the resulting inward-open model structure. In this model, the extracellular gate is closed by the side-by-side interactions between TM2 and TM8. The conserved Gly277 residue on TM7 interacted with TM4 in the N-lobe, as observed for the Gly residues in the intracellular gate. In addition, Asp139 and Asn281, as well as Ala67 and Ser290, are located near each other in the model structure, suggesting that these residue pairs are involved in the extracellular gate formation (Fig. 3e, f). To verify the putative interactions in the extracellular gate, we calculated the evolutionary coupling pairs from the 36,873 MATE family members, using the program EVfold_membrane[28,29]. The high EC values for the predicted residue pairs in the putative

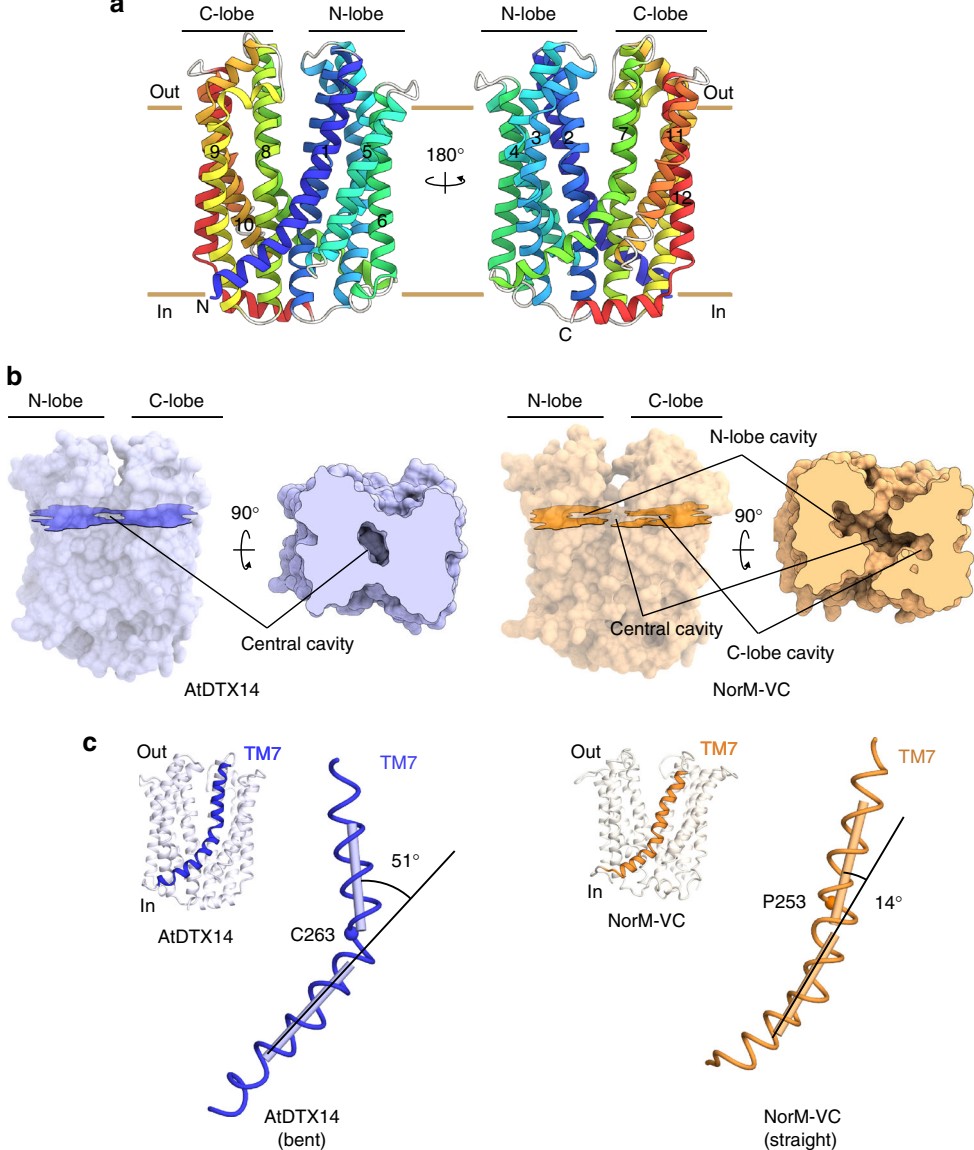

**Fig. 2** Overall structure of AtDTX14. **a** Cartoon representations of the crystal structure of AtDTX14, viewed parallel to the membrane. The structures are colored from blue to red from the N terminus to the C terminus. **b** Surface representations and cross-sections of AtDTX14 and NorM-VC (PDB ID: 3MKT). The cross-sections are viewed perpendicular to the membrane, for comparison of their cavities. **c** Comparison of the helix axis angles of TM7 between AtDTX14 (blue) and NorM-VC (orange). All molecular graphics were created with the program CueMol (http://www.cuemol.org/)

extracellular gate indicated that these residue pairs have co-evolved to allow the extracellular gate formation in the MATE family transporters. The calculation revealed that the residue pairs in the gate (Ala67-Ser290, Asp139-Asn281, and Gln138-Gly277) have high EC scores, supporting the validity of the interactions predicted by our inward-open model structure (Fig. 3e, f; Supplementary Fig. 5).

**C-lobe structure and hydrogen-bonding network**. Although the structures of the N-lobe and C-lobe of AtDTX14 are similar to those of the bacterial MATE transporters, respectively, TM7 in the C-lobe of AtDTX14 assumes a bent conformation. This is the most distinct point from the other reported MATE structures, as TM7 is bent by about 50° around Cys263 as a hinge region (Fig. 2c). Interestingly, an extensive hydrogen-bonding network is formed around this hinge region, involving Glu265, Gln296, Asp383, Asn406, Gln443, and Tyr410 (Fig. 4). In this hydrogen-

bonding network, Glu265, Asp383, Asn406, Gln443, and Tyr410 are well conserved in the eMATE subfamily (Supplementary Fig. 1), suggesting the existence of a similar hydrogen-bonding network in other eMATE transporters. In the present crystal structure of AtDTX14, Glu265, residing near the hinge region of TM7, hydrogen bonds with Asp383 on TM10 (Fig. 4). Given the crystallization conditions of AtDTX14 (pH 5), Glu265 or Asp383 is probably protonated to form this hydrogen bond.

To further explore the importance of this hydrogen-bonding network in the C-lobe, we created AtDTX14 mutants and performed functional analyses, using the complementation assay with the *E. coli* 6-KO strain. We also verified the structural integrity of the mutants by fluorescence-based size-exclusion chromatography[25] (Supplementary Fig. 6). The results showed that the Q296A and Q443A mutants have decreased complementation activities (Fig. 1), suggesting the importance of this hydrogen-bonding network for the transport mechanism. Furthermore, the D383A mutant also exhibited decreased

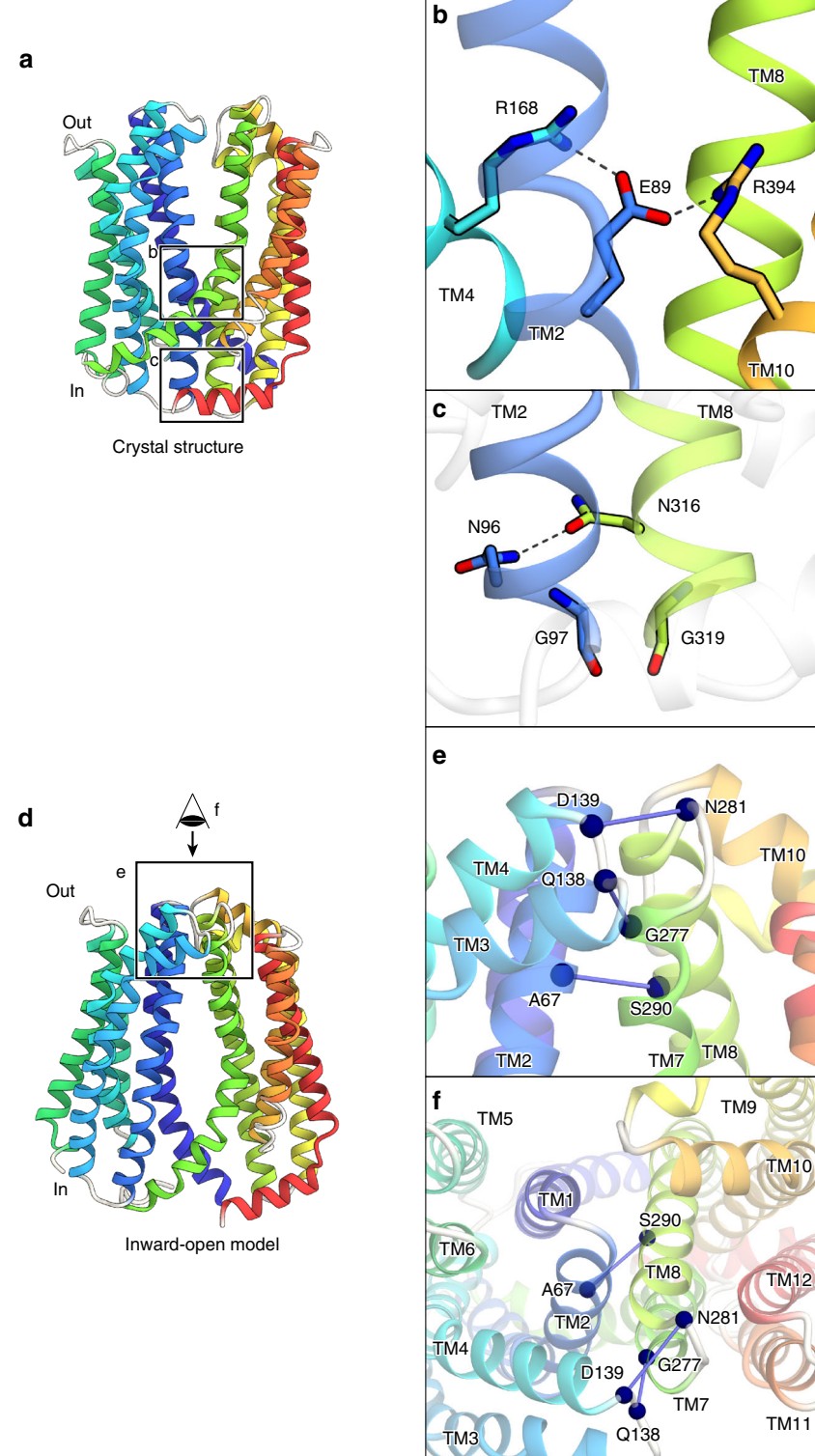

**Fig. 3** Internal cavity, intracellular gate, and predicted extracellular gate. Crystal structure of AtDTX14 (**a–c**) and the modeled structure of the inward-open state (**d–f**), viewed parallel to the membrane. The model structure was created by, respectively, superimposing the N-lobe and C-lobe of AtDTX14 onto the N-lobe and C-lobe of the crystal structure of MurJ (PDB ID: 5T77), using the SSM superpose algorithm[41]. The resulting RMSD values are 2.90 Å and 2.06 Å for the N-lobe and C-lobe, respectively, indicating their structural similarities despite the low-sequence identity (~4%). **b** Close-up view of the bottom of the internal cavity. The bottom is closed by the salt-bridge network, which is represented by dashed lines. **c** Close-up view of the intracellular gate. The intracellular Gly pair (G97 and G319) enables the side-by-side interaction between TM2 and TM8. **e**, **f** Close-up views of the putative extracellular gate, viewed parallel to the membrane (**e**) and perpendicular to the membrane (**f**). The blue spheres and lines indicate the residue pairs with high EC scores, calculated by the program EVfold_membrane[28,29]

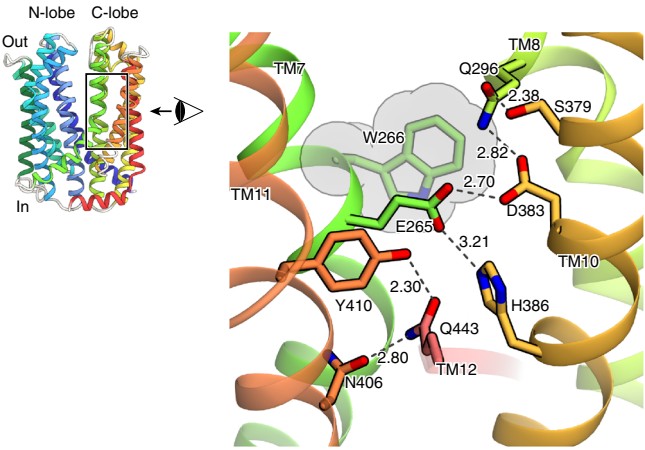

**Fig. 4** Hydrogen-bonding network in the C-lobe. Close-up view of the C-lobe hydrogen-bonding network is shown. The dashed lines indicate the hydrogen-bonding network. Trp266 is represented by a CPK model (gray)

complementation activity (Fig. 1). This result suggests that the protonation and the hydrogen bonding between the conserved acidic residues play pivotal roles for xenobiotic extrusion by AtDTX14. Proton binding at these conserved acidic residues is likely to have a key role in coupling the proton-motive force to the substrate extrusion in eMATE.

One of the interesting points about this hydrogen-bonding network is that it is occluded from the central cavity, as well as from the extracellular space, by the hydrophobic barrier (Supplementary Fig. 7). In the AtDTX14 structure, this hydrophobic barrier consists of conserved hydrophobic residues including Trp266, which is well conserved in the eMATE subfamily (Supplementary Fig. 1). To investigate the importance of this Trp266 residue, we performed the complementation assay of the W266A mutant of AtDTX14. The complementation activity was slightly decreased (Fig. 1), suggesting that the occlusion of this hydrogen-bonding network from the solvent is important for the transport activity. The occlusion of the conserved acidic residues in the low dielectric environment by Trp266 may increase their pKa values, thereby facilitating proton binding to this site.

**Insight into human MATE transporters**. The amino-acid sequence identity between hMATE1 and AtDTX14 is as high as 32%, which enabled us to create a more accurate model of the hMATE1 structure, based on the present crystal structure of AtDTX14. This model structure suggested that the hydrogen-bonding network in the C-lobe, including the hydrogen bonding between the conserved acidic residues, is also conserved in hMATE1 (Fig. 5a). To investigate the importance of this hydrogen-bonding network, we created hMATE1 mutants and analyzed their functions. We measured the uptake activity of radioisotope-labeled tetraethylammonium (TEA) or cimetidine into human embryonic kidney 293 (HEK-293) cells expressing hMATE (and its mutants) in high-pH buffer, as in the previous study[3]. We also verified that all of the mutants are expressed and localized correctly within the plasma membrane of HEK-293 cells (Supplementary Figs 8, 9, 13). The structural integrities of these mutant proteins were verified by the fluorescence-based size-exclusion chromatography analysis of the corresponding mutants of AtDTX14 (Supplementary Fig. 6). The results showed that the mutations of the conserved acidic residues, Glu273 (Glu265) and Glu389 (Asp383), reduced both the TEA and cimetidine transport activities (AtDTX14 numbering is indicated in parentheses) (Fig. 5b, c), supporting our notion that this site functions as the

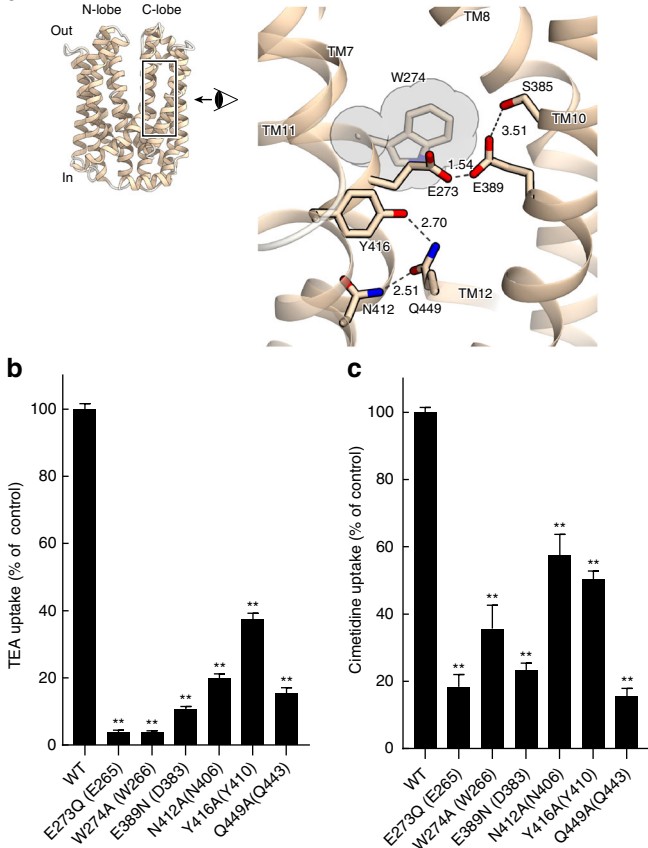

**Fig. 5** Homology model of hMATE1 and transport assay. **a** Homology model of hMATE1 and close-up view of the hydrogen-bonding network in the C-lobe. The model structure was generated by the program MODELLER[42], using the AtDTX14 structure as the template. The residues involved in the hydrogen-bonding network were manually adjusted, using the program COOT[38]. In the close-up view, Trp274 is represented by a CPK model (gray). **b**, **c** Uptake of [1-¹⁴C]-labeled TEA (**b**), and [N-methyl-³H]-labeled cimetidine (**c**) into HEK-293 cells expressing hMATE1 mutants. Activities relative to that of the wild type (WT) are shown. The control activity (100%) of TEA and cimetidine correspond to $6 \pm 0.17$ nmol mg⁻¹ protein and $47 \pm 2.7$ pmol mg⁻¹ protein, respectively. The labels on the horizontal axis indicate the mutated amino-acid residues of each hMATE1-mutant (AtDTX14 numbering is indicated in parentheses), $n = 8$–33. Data are means $\pm$ SEM, $**P < 0.01$ (two-tailed paired Student's $t$ test)

proton binding site. This result is also consistent with a previous functional analysis of hMATE1, showing that Glu273 and Glu389 are critical for the transport activity[30]. Furthermore, we analyzed the transport activities of the other residues in this hydrogen-bonding network. The results showed that the mutation of Tyr416 still retains ~50% activities for both TEA and cimetidine (Fig. 5b, c), indicating that this Tyr residue does not have a critical role in the substrate transport mechanism. In contrast, the mutation of Gln449 (Gln443) affected both the TEA and cimetidine transport activities, and the mutation of Asn412 (Asn406) strongly reduced the TEA transport activity (Fig. 5b, c). These results suggested that Asn412 is directly involved in the recognition of TEA and TEA-like substrates (Fig. 5b, c). A previous functional analysis of hMATE1 revealed that the Asp mutant of Glu273 (Glu265) exhibited a different substrate specificity from that of the wild-type transporter[30]. Taken together, these observations suggest that the residues in this hydrogen-bonding network in the C-lobe function as the

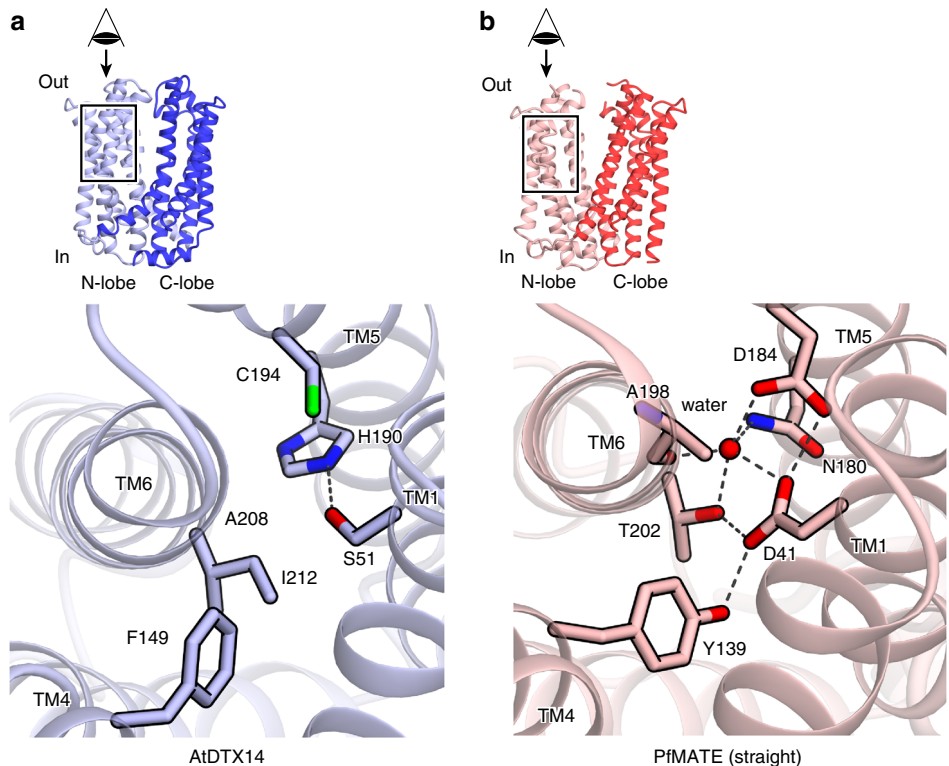

**Fig. 6** Structural comparison between the N-lobes of AtDTX14 and PfMATE. **a**, **b** Close-up views of the N-lobe of AtDTX14 (**a**) and that of PfMATE (**b**)

substrate-binding site. Next, we created the Ala mutant of Trp274 (Trp266) in hMATE1 (Fig. 5a), which occludes the hydrogen-bonding network in the present crystal structure (Fig. 4), and analyzed its transport activity. This W274A mutation affected both the TEA and cimetidine transport activities, suggesting that Trp274 in hMATE1 also functions to occlude the hydrogen-bonding network.

**Structural comparison with bacterial MATE subfamilies**. In the present crystal structure of AtDTX14, the aspartate residue in TM1 that is critical for the transport activity in the DinF subfamilies is not conserved, and is replaced with more compact residue that is incapable of proton binding (Fig. 6; Supplementary Fig. 1). The other residues involved in the hydrogen-bonding network in the N-lobes of the DinF subfamily members are also not conserved in eMATE (Fig. 6; Supplementary Fig. 1). Thus, the present structure suggests that the eMATE subfamily does not utilize a transport mechanism based on the structural change of TM1, which is likely to be conserved in the DinF subfamily[20]. The transport mechanism of eMATE may be different from that of DinF, and the N-lobe of eMATE plays a distinct role from those of the bacterial MATE transporters.

We next compared the C-lobe structure of eMATE with those of the bacterial MATE transporters. The conserved acidic residues in the C-lobe of eMATE (Glu265 and Asp383 in AtDTX14) are not present in the C-lobes of the DinF subfamily members, suggesting that the role of the C-lobe is also different. In contrast, these C-lobe acidic residues are also conserved in the NorM subfamily (Glu255 and Asp371 in NorM-VC; Supplementary Fig. 1), and were shown to be critical for the transport activity[19,21,24]. Thus, the C-lobes of both eMATE and NorM may share an important role in the substrate transport mechanism[23]. Despite this similarity at the amino-acid sequence level, we found significant differences between the C-lobe structures of eMATE and NorM. In the crystal structure of

NorM-VC, TM7 assumes a straight conformation and the helix axis angle between the extracellular and intracellular halves of TM7 is ~15°, in contrast to the corresponding angle in the AtDTX14 structure (~50°) (Fig. 2c). As a result, the conserved acidic residues (Glu255 and Asp371) do not interact, and are separated by about 8.9 Å (Fig. 7b). A large cavity (C-lobe cavity) is formed between TM7 and TM10, and these acidic residues are exposed to this C-lobe cavity (Fig. 7b). Given the neutral pH of the NorM-VC crystallization conditions (pH 7.2–8.6), it is likely that both Glu255 and Asp371 are ionized. The electrostatic repulsion between these two acidic residues may force them apart in the NorM-VC structure, and stabilize the straight conformation of TM7. Furthermore, given the sequence similarity between the C-lobes of eMATE and NorM, it is quite likely that the C-lobe of eMATE also assumes a similar conformation to this NorM-VC structure under high-pH conditions, in which the conserved acidic residues are ionized and TM7 adopts the straight conformation.

## Discussion

The structural comparison between AtDTX14 and NorM leads to a plausible hypothesis for the eMATE subfamily: the protonation of the conserved acidic residue in the C-lobe (Glu265 or Asp383 in AtDTX14) relaxes the electrostatic repulsion and enables the hydrogen-bond formation between them, resulting in the straight to bent conformational change of TM7 (Fig. 8). In the outward-open state with the straight TM7 conformation, the C-lobe cavity is formed among TM7, TM8, and TM10, and is suitable for the recognition of the positively charged substrate (Fig. 8a; Supplementary Fig. 10a). In this straight conformation, the hydrogen-bonding network in the C-lobe is disrupted and the residues involved in the network are exposed to the C-lobe cavity to interact with the substrate. After the proton binding to Glu265/Asp383, the structural change of TM7 from the straight to bent conformation collapses this C-lobe cavity and enables the

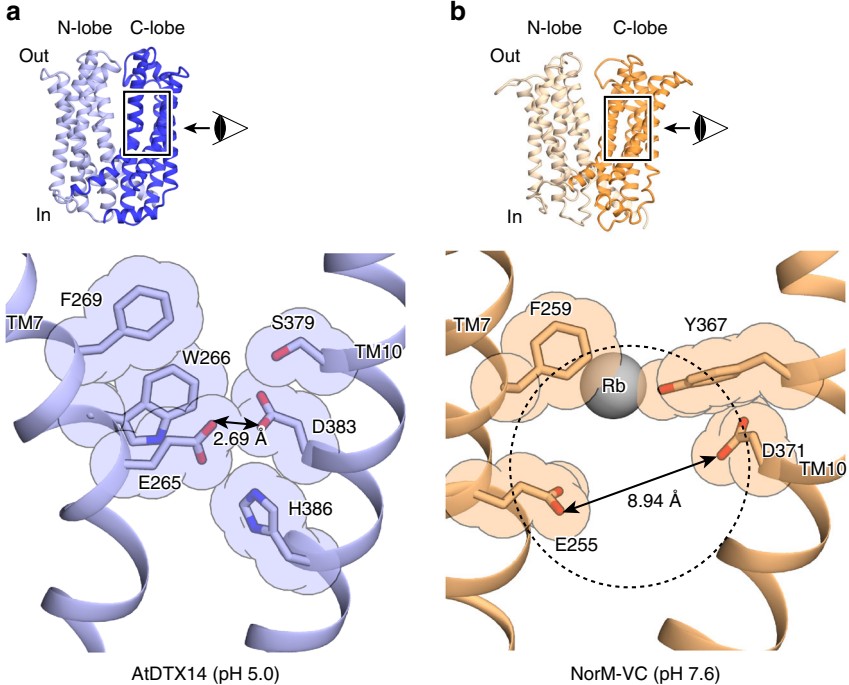

**Fig. 7** Structural comparison of the C-lobes of AtDTX14 and NorM-VC. **a**, **b** Close-up views of the putative substrate pocket of AtDTX14 (**a**) and that of NorM-VC (**b**) located at the C-lobe. The gray sphere represents a rubidium ion, and the dashed circle indicates the putative substrate pocket of NorM-VC. The double-headed arrows indicate the distances between the glutamate and aspartate residues, which are essential for the transport activity and the substrate recognition

formation of the hydrogen-bonding network observed in the present crystal structure (Fig. 8b). This collapse of the C-lobe cavity may be important to prevent the re-binding of the substrate from the extracellular side, thereby ensuring the unidirectional (uphill) movement of the substrate across the membrane. In the course of the structural change of TM7, the conformational change of Trp266 sequesters Glu265/Asp383, to facilitate the protonation of Glu265/Asp383 (Fig. 8a, b). Furthermore, this coupling mechanism between TM7 bending and proton binding can explain the exclusivity of the substrate and proton binding, which underlies the antiporter mechanism. In the proton-bound state, TM7 in the bent conformation collapses the substrate-binding pocket, and the substrate cannot bind to the transporter, whereas in the proton-unbound state, the straight conformation of TM7 creates a suitable binding site for the accommodation of positively charged substrates (Supplementary Fig. 10a).

To examine the effect of the structural conversion from the outward-open to inward-open states on the C-lobe cavity, we created the inward-open model structure with the straight TM7 conformation based on the MurJ structure (Supplementary Fig. 10b), using the same method as that for the bent TM7 conformation. The model showed that the C-lobe cavity is also conserved in the inward-open model structure, and its accessibility to the intracellular and extracellular spaces is controlled by the rocking motion of the N-lobe and C-lobe, which opens and closes the extracellular and intracellular gates (Supplementary Fig. 10). Thus, this C-lobe cavity is a plausible substrate-binding pocket for a transporter that operates by the rocking bundle mechanism[31,32].

The structural comparison between AtDTX14 and NorM also suggested a possible transport mechanism for the NorM subfamily. The co-substrate (i.e., H$^+$ or Na$^+$) binding to the conserved acidic residues in the C-lobe (Glu255 and Asp371 in NorM-VC) may cause the straight to bent conformational change of TM7, which collapses the C-lobe cavity. In the case of the

Na$^+$-driven NorM transporters, (e.g., NorM-VC and NorM-NG), Na$^+$-binding to the C-lobe may mediate the interaction between Glu255 and Asp371, thereby inducing the kink of TM7 and collapsing the C-lobe cavity. A previous biophysical analysis of *Pseudomonas stutzeri* NorM-PS revealed that the conserved acidic residues in the C-lobe are directly involved in the substrate recognition[33], suggesting that the C-lobe cavity functions as a substrate-binding site, as in the eMATE mechanism. In contrast, several crystal structures of NorM-NG in complex with organic cations were reported[21,23], in which the binding sites are different from the C-lobe cavity proposed in this study (Supplementary Fig. 11). These organic cations are bound to the site between the extracellular halves of the N-lobe and C-lobe, where the extracellular gate is formed in the inward-open conformation. If this NorM-NG site were the substrate-binding site, then the inward-open structure would be impossible, because the N-lobe and C-lobe would clash with the bound substrate. Therefore, it is highly likely that these organic cations in the NorM-NG crystal structures are transiently trapped in a minor site in the transport pathway from the C-lobe cavity to the extracellular space.

It is interesting to note that, in the DinF-BH structure, both TM7 and TM8 are bent and the extracellular half of the TM7-TM8 hairpin contacts the N-lobe. As a result, a wide crevice is formed by the extracellular halves of the C-lobe, which was hypothesized to be important for the transport mechanism of DinF-BH[22]. Although this structural change also involves the kink in TM7, the resulting bent structure of DinF-BH is quite different from the bent conformation of AtDTX14 (Supplementary Fig. 12). Moreover, this structural change was suggested to be a step in the transition between the occluded and outward-open states, which are not coupled to the substrate/co-substrate binding[22]. Therefore, this bending of the TM7-TM8 hairpin in DinF-BH is substantially different from the structural change in the C-lobe proposed for the eMATE transporters in the present work. This structural difference also further accentuates the

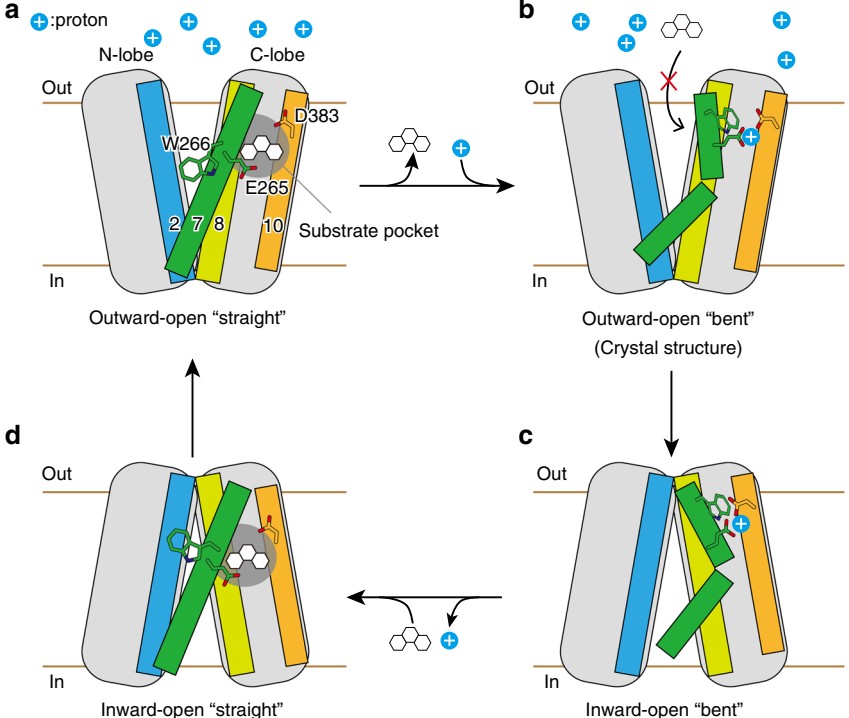

**Fig. 8** Proposed transport mechanism of the eMATE transporters. Schematic diagram illustrates the possible transport cycle of the eukaryotic MATE transporters. The key residues involved in the substrate recognition and the conformational change are shown

differences in the C-lobe functions between the eMATE and DinF transporters.

In conclusion, the present crystal structure of AtDTX14 suggested the transport mechanism by the structural change of TM7 in the C-lobe, which may be commonly conserved in the eMATE subfamily, including hMATE1. This mechanism differs from the transport mechanism of the DinF subfamily members, which involves the structural change of TM1 in the N-lobe. Further structural studies, including the determination of the substrate-bound structure and/or the outward-open structure of eMATE, will be required to completely understand the transport mechanism of eMATE.

## Methods

**Expression and purification**. The AtDTX14 gene (UniProt ID: Q9C994, NCBI Reference Sequence: NP_177270.1, Supplementary Table 1) was artificially synthesized (GenScript). For crystallization, the N-terminal and C-terminal flexible regions (residues 1–19 and residues 473–485) were truncated, and Pro36 was mutated to Ala to facilitate crystallization. The resulting gene was spliced into the pFastBac1 expression vector, which contains the *EGFP* gene following the TEV protease cleavage site. This construct of AtDTX14 was expressed in Sf9 cells (ATCC CRL-1711[TM]), using the Bac-to-Bac baculovirus expression system (Invitrogen). Sf9 cells were infected at a density of $2.5$–$3 \times 10^6$ cells ml$^{-1}$, and the infected Sf9 cells were cultured in Sf900II medium (Invitrogen) at 27 °C for 60 h. The cells were collected by centrifugation at $6000 \times g$ for 10 min. The pellets were disrupted by sonication in 50 mM Tris-HCl (pH 8), 150 mM NaCl, and 0.1 mM PMSF. The cell debris was removed by centrifugation at $10,000 \times g$ for 30 min, and the membrane fraction was collected by ultracentrifugation ($125,000 \times g$ for 1 h). This fraction was solubilized in buffer containing 150 mM NaCl, 50 mM Tris-HCl (pH 8), 0.1 mM PMSF, 1.5% *n*-dodecyl-β-D-maltoside (DDM), and 0.3% cholesteryl hemisuccinate (CHS). The insoluble fraction was removed by ultracentrifugation ($125,000 \times g$ for 30 min), and the supernatant was mixed with Ni-NTA resin (QIAGEN). After binding for 1 h, AtDTX14 was eluted in buffer containing 150 mM NaCl, 50 mM Tris-HCl (pH 8), 400 mM imidazole, 0.05% DDM, and 0.01% CHS. The N-terminal EGFP-His$_8$ tag was cleaved by His-tagged TEV protease (made in-house), and the sample was reloaded onto the Ni-NTA column to remove the cleaved EGFP-His$_8$ tag and the protease. The flow-through containing AtDTX14 was collected, concentrated, and further purified by size-exclusion chromatography in 150 mM NaCl, 50 mM Tris-HCl (pH 8), 0.05% DDM and 0.01% CHS. Peak fractions were collected and concentrated to ~40 mg ml$^{-1}$ for crystallization. Before crystallization, AtDTX14 was incubated with 0.2 mM cimetidine for 1 h and ultracentrifuged at $86,400 \times g$ for 20 min.

**Crystallization**. AtDTX14 was mixed with monoolein (Sigma) in a 4:6 protein to lipid ratio (w/w), and reconstituted in the lipidic cubic phase (LCP) using coupled syringe devices. A 40 nl portion of the protein–LCP solution mixture was dispensed on a 96-well sandwich plate and overlaid with 700 nl of precipitant solution by the crystallization robot, Crystal Gryphon LCP (Art Robbins Instruments). The best crystals were obtained from the precipitant solution containing 100 mM Na-citrate, pH 5–5.3, 100 mM MgSO$_4$, 50–100 mM NaK-tartrate-tetrahydrate and 20–24% PEG550MME. The crystals were incubated for 10–14 days at 20 °C, harvested using mesh grid loops (MiTeGen), and flash cooled in liquid nitrogen.

**Data collection and structure determination**. All diffraction datasets were collected at SPring-8 BL32XU using MX225HS CCD detector (Rayonix, LLC). Datasets were acquired at 100 K using a beam size of $10 \times 10$–$18 \times 10$ μm$^2$ with 1 Å wavelength X-rays. The data collection was performed manually or automatically. The automatic data collection was performed using the ZOO system. In both manual and automatic data collection, the loop-harvested microcrystals were identified by raster scanning and analysis using the program SHIKA[34]. Small wedge (5–20°) data were collected from single crystals and the datasets were processed automatically using the program KAMO (https://github.com/keitaroyam/yamtbx). For indexing and integrating of datasets, the XDS[35] was used. The datasets were hierarchically clustered by using the correlation coefficients of the intensities between datasets. Finally, the outlier-rejected datasets group was scaled and merged using XSCALE[35]. The initial phases were obtained by molecular replacement method with the program MOLREP[36] using poly-alanine model of NorM-VC (PDB ID:3MKT) as the search model. The solution of the molecular replacement was refined using the program REFMAC[37], including the 100-iteration jelly-body refinement. Then, the structural model was manually rebuilt using COOT[38] and was refined using the program PHENIX[39]. The data collection and refinement statistics are summarized in Table 1. All molecular graphics were illustrated using CueMol (http://www.cuemol.org/).

***In vivo* complementation analysis**. The drug-sensitive strain (BW25113 *acrAB::Δ macAB::Δ yojHI::Δ*) was transformed with each IPTG inducible plasmid, i.e., IPTG inducible vector pSUIQ[20] and ones harboring wild-type or mutant AtDTX14 cDNAs, and the transformants were obtained primarily on LB agar plates (7 μg ml$^{-1}$ chloramphenicol, 0.5% glucose). Each colony was grown in LB liquid media (7 μg ml$^{-1}$ chloramphenicol, 0.5% glucose) to log-phase (OD$_{600}$ < 0.60, ~$5 \times 10^8$ cells ml$^{-1}$) and the cell density of each culture was correctly adjusted to OD$_{600}$ = 0.5. Then, 1/50 diluted solutions of each adjusted culture (6 μl)

was spotted on LB plates (7 µg ml$^{-1}$ chloramphenicol) with or without 0.02 µg ml$^{-1}$ norfloxacin and 0.25 mM IPTG, and cell growth was monitored at 12 to 18 h.

**Fluorescence-based size-exclusion chromatography analysis**. All AtDTX14 mutants were spliced into the pNGFP_EU expression vector, which contains the EGFP gene. These constructs were transfected into HEK-293T cells using the Lipofectamine 2000 reagent (Invitrogen), for transient expression. HEK-293T cells were transfected at a density of $1 \times 10^6$ cells ml$^{-1}$, and the transfected HEK-293T cells were cultured in Dulbecco's Modified Eagle's Medium (SIGMA-ALDRICH) at 37 °C for 36 h under 5% CO$_2$. The cells were collected by centrifugation at 6000×g for 3 min. The cell pellets were solubilized in buffer containing 150 mM NaCl, 50 mM Tris-HCl (pH 8), 0.1 mM PMSF, 1.5% n-dodecyl-β-D-maltoside (DDM), and 0.3% cholesteryl hemisuccinate (CHS). The insoluble fraction was removed by ultracentrifugation (125,000×g for 20 min). The supernatants were loaded onto an ENrich SEC650 column (Bio-Rad) equilibrated with running buffer (150 mM NaCl, 50 mM Tris-HCl (pH 8), 0.05% DDM, and 0.01% CHS), and the proteins were detected by the EGFP fluorescence, with excitation at 480 nm and emission monitored at 512 nm.

**HEK-293 cell-based transport assay**. HEK-293 cells ($0.8 \times 10^5$ cells per well in 24-well plate) were grown in DMEM containing 10% FBS, 100 µg ml$^{-1}$ penicillin, and 0.25 µg ml$^{-1}$ streptomycin, at 37 °C under 5% CO$_2$ atmosphere. After incubation for 24 h, 0.5 µg of pcDNA3.1/hMATE1-WT, pcDNA3.1/hMATE1-mutant, or pcDNA3.1 was transfected into HEK-293 cells by lipofection, using the TransIT reagent (Mirus). After further incubation for 2 days, the cells were washed twice with transport assay medium containing 125 mM NaCl, 4.8 mM KCl, 5.6 mM D-glucose, 1.2 mM CaCl$_2$, 1.2 mM KH$_2$PO$_4$, 1.2 mM MgSO$_4$, and 25 mM Tricine (pH 8). The cells were incubated in transport assay medium containing 50 µM [1-$^{14}$C] TEA (0.5 MBq µmol$^{-1}$) (3.7 kBq per assay; Perkin Elmer Life Science) or 1 µM [N-methyl-$^3$H] cimetidine (18.5 MBq µmol$^{-1}$) (American Radiolabeled Chemicals Inc.) at 37 °C for 20 min, and were washed twice with 500 µl ice-cold assay medium. The cells were lysed with 400 µl of 1% SDS, and the radioactivity was measured by liquid scintillation counting (Perkin Elmer).

**Immunohistochemistry**. The cells on poly-L-lysine-coated coverslips were fixed with 4% paraformaldehyde in phosphate-buffered saline (PBS) for 30 min at room temperature. The specimens were washed with PBS, and were incubated for 20 min in PBS containing 0.1% Triton X-100. The specimens were incubated with PBS containing 2% goat serum and 0.5% bovine serum albumin for 20 min, and were incubated with anti-hMATE1 antiserum (1:1000 dilution with PBS containing 0.5% bovine serum albumin) for 1 h at room temperature. After four washes with PBS, the specimens were incubated with Alexa Fluor488-labeled anti-rabbit IgG (2 µg ml$^{-1}$) for 1 h at room temperature. After seven washes with PBS, the specimens were mounted with Fluoromount (Diagnostic Biosystems), and were observed using an Olympus BX60 microscope.

**Western blot**. Total proteins from HEK-293 cells expressing WT and mutant hMATE1 (20 µg) were subjected to electrophoresis on a 10% polyacrylamide gel. After the proteins were transferred to a nitrocellulose membrane and incubated with the anti-hMATE1 antibody, the immunoreactive proteins were visualized using an ECL detection kit (GE Healthcare). The protein concentration was determined using bovine serum albumin as the standard[40].

**Data availability**. The structure factors and coordinates of AtDTX14 (P36A mutant) were deposited to Protein Data Bank (PDB ID: 5Y50). Other data are available from the corresponding authors upon reasonable request.

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

## Acknowledgements

We thank H. Nishimasu (University of Tokyo, Japan) for useful discussions, the beamline staff at BL32XU of SPring-8 (Hyogo, Japan) for technical support with data collection, and A. Kurabayashi for technical assistance. The X-ray diffraction experiments were performed at SPring-8 BL32XU (proposal Nos. 2015A1024, 2015B2024, 2016A2527, and 2016B2527) with the approval of RIKEN. This work was supported by the Platform for Drug Discovery, Informatics, and Structural Life Science, funded by the Ministry of Education, Culture, Sports, Science and Technology (MEXT), and by a Grant-in-Aid for Specially Promoted Research (16H06294), and a Grant-in-Aid for Scientific Research (B) (25291011) from the Japan Society for the Promotion of Science (JSPS) to O.N. and R.I., respectively.

## Author contributions

H.M. purified and crystallized AtDTX14, and collected the diffraction data. H.M. solved the structure. T.K. and N.D. assisted with the construction design, purification, and crystallization. K.Y., K.H., Nakane T., Nishizawa T. and K.K. performed data processing. K.K. and T.K. supported the modeling and refinement. H.M. made the mutants for transport analyses. S.M., T.M. and Y.M. performed the transport analyses using HEK-293 cells. K.I. performed the in vivo complementation assays. M.H., T.M., T.K., R.I. and O.N. designed the research, and M.H., T.M., R.I. and O.N. wrote the manuscript. R.I. and O.N. directed and supervised all of the research.

## Additional information

**Competing interests:** The authors declare no competing financial interests.

