## [Peer Review file · Nature Communications]

Reviewers' comments:

Reviewer #1 (Remarks to the Author):

This manuscript presents a high resolution structure of the first eukaryotic MATE (eMATE) transporter, AtDTX14 from *Arabidopsis thaliana*. As for most transporters of the MATE family, well diffracting crystals were obtained in cubo. In fact, a mutation of a highly conserved proline in transmembrane helix 1 (TM1) into alanine was required to obtain the crystals. The structure is compared to various other outward-facing bacterial MATE transporters, belonging to the DinF and NorM subfamily, respectively. The major finding of the study is the presence of a conserved hydrogen bonding network within the C-lobe of AtDTX14, shielded by a Trp-residue, which is reminiscent of a similar network found in the C-lobe of the NorM subfamily and the symmetry-related N-lobe of the DinF family. In members of the DinF family, a corresponding hydrogen bonding network in the N-lobe had been postulated to be responsible for bending movements in TM1 as a consequence of protonation/deprotonation of conserved acidic residues and thereby driving substrate transport. By comparing structures of AtDTX14 with the NorM subfamily member NorM-VC, an analogous bending movement in the symmetry related TM7 was observed. While transport of novobiocin by AtDTX14 was demonstrated in *E. coli*, the transport activity appeared too low for conducting functional assays with AtDTX14. Therefore, the authors resorted to human hMATE1 for their functional studies, for which a robust *in vivo* transport assay of tetraethylammonium (TEA) exists and interrogated residues of the hydrogen bonding network of the C-lobe as well as glycine residues places at the intra- and extracellular gates closed by the two lobes when the transporter adopts its inward- as well as its outward-facing state. From a structural biology point of view and given the high resolution obtained, the work is excellent and provides novel insights into a new subfamily of MATE transporters, the eMATEs. From a functional point of view, the authors provide some evidence for their hypotheses. However, these are solely based on a single type of *in vivo* transport assays conducted with a homologous transporter. While the quality of the figures is excellent, the manuscript is somewhat more difficult to digest, presumably due to the quite lengthy structural descriptions, which are not that attractive to read. Nevertheless, the manuscript is clearly written.

Major points

1) The authors compare the AtDTX14 structure with NorM-VC and draw conclusions regarding the functional importance of TM7 bending. Therefore it seems that eMATEs are more closely related to the NorM subfamily of MATE transporters as opposed to the DinF subfamily. A phylogenetic tree including members of the three MATE subfamilies would be very helpful to clarify this question. Should the eMATEs not be that closely related to the NorM subfamily, I consider it as not entirely adequate to draw mechanistic conclusion from a comparison between structures of AtDTX14 and NorM-VC.

2) All MATEs have so far been crystallized in outward-facing states with a closed intracellular gate. Recently a structure inward-facing MurJ (belonging as the MATEs to the MOP superfamily of transporters) was described, based on which the authors made a homology model. At both the intra- and extracellular gate the authors spotted conserved glycine residues which were mutated into bulky glutamine residues in hMATE1. The corresponding mutants were almost inactive in terms of transport. However, replacing a conserved glycine by a bulky glutamine likely results in major structural disturbances. The authors at least have to show whether the respective mutants (introduced in AtDTX14) can still be purified and behave the same as the wildtype protein on size exclusion chromatography analysis.

3) The inward-facing homology model was validated by calculating the evolutionary coupling pairs. However, it should in addition be validated by strategically placed cysteine pairs, which only can form a disulfide bond when the transporter adopts its presumed inward-facing state.

4) Supplementary Figure 2: The novobiocin resistance conferred by AtDTX14 in *E. coli* seems to be very weak. In addition, the picture is only partially self-explanatory and difficult to understand even with the legend. I had difficulties to understand that the numbers provided in the right panel correspond to the constructs shown on the left panel. Further, I do not understand what can be

learned from the left panel. The strains contain the respective constructs on a plasmid and grow fine in the absence of inducer IPTG and novobiocin, i.e. I do not see any insight one can gain from this. Finally, I would see it as mandatory to introduce mutations into AtDTX14, which abrogate its function (e.g. E265Q or D383N) to validate the results obtained in the presence of novobiocin + IPTG as shown on the right panel at the bottom.

5) Concerning the interaction between E265 and D383: this may in fact represent a low-barrier hydrogen bond. According to figure 5, the distance is 2.69 Å, which rather points towards a classical hydrogen bond. This interaction may deserve a bit of deeper thinking. With respect to the model in Figure 6, it is important to indicate which state corresponds to the crystal structure (outward-open "bent"). Further, it would be nice to have a depiction of the conserved Trp residue, which shields the collapsed C-lobe cavity in the outward-open "bent" structure.

Minor points

1) Concerning the mutations in the hydrogen bonding network, in particular E265 and D383 (AtDTX14 numbering), it is interesting that the authors did exchange the residue into Gln and Asn, respectively. This was a very good choice, because in this way the mutation only removed the protonatable group without changing the side chain length. Perhaps the authors should mention that in the previous publication they cite (Ref 25), the respective residues were exchanged for aspartates and alanines.

2) Concerning the pK of the acidic residues within the C-lobe, it may be worth mentioning that in a hydrophobic environment, the aspartates and glutamates may in fact exhibit pKa in the range of 6-7. This has been shown for many proton translocating transporters including the F1Fo-ATPase and RND pumps such as AcrB. By removing the conserved Trp residues, the pKa values of E265 and D383 may be much decreased, thereby explaining the loss of function.

Reviewer #2 (Remarks to the Author):

The manuscript, "Structural basis for xenobiotic extrusion by eukaryotic MATE transporters" reports the X-ray structure of the eukaryotic MATE transporter, AtDTX14 (AtDTX), from *Arabidopsis thaliana* at 2.6Å;. The authors employ structure-based comparisons of AtDTX with MATE proteins belonging to other subfamilies, including NorM, DinF and MurJ to guide structure-function studies of a human homolog hMATE, identify potentially important functional-structural determinants and ultimately propose possible a mechanism for xenobiotic recognition and transport by AtDTX.

Importantly, this appears to be the first X-ray structure of a protein from the eMATE family. Structural departures from the bacterial MATE proteins and their mechanistic consequences could also be important if understood at the molecular level. The manuscript aims to achieve such a description of both structure and mechanism.

Unfortunately, the manuscript does not at all read very clearly nor does it follow a clear logical narrative. Many of the details of the AtDTX structure are lost in the very detailed (sometimes-- maybe inappropriate) comparisons with MATE proteins belonging to other subfamilies. The study is also weakened by the fact that mutational studies are not used to probe the reported AtDTX structure. Instead, they are used to probe a homologous protein, hMATE. The reviewer feels that the minimal set of reported mutational data with the single AtDTX structure are not sufficient to formulate the mechanism proposed for transport (see discussion below). Additional structures,

including a substrate-bound AtDTX as well as mutational studies on the plant system might afford a more compelling manuscript. Finally, the mechanistic model proposed for xenobiotic transport is very similar to that proposed for DinF (Lu (2013) NSB 20, 1310). The DinF transport mechanism also includes structural changes in TM7 (see manuscript abstract). As such, the novelty of the results and mechanistic proposal are questionable.

(page 4, paragraph 2) The authors claimed to have identified the cation binding pocket in AtDTX based on the (1) presence of two acidic residues (called a conserved acidic cluster) and (2) binding of Rb⁺ and Cs⁺ ions to NorM in a cavity lined by two homologous acidic residues. However, ligand-bound structures of NorM do not show lipophilic cations bound to the same region as the Rb⁺ and Cs⁺ ions (pdb ids: 4huk (P4P), 4hum (ET), 4hun (RHQ), 4hul (Cs), 3mku (Rb) (NorM) as well as 4lz6, 4lz9 (RHQ), 3vvp (Br-NRF) (DinF). Also, an analysis of the RH6G-bound DinF structure does not show an acidic cluster near RH6G. Moreover, a Br-NRF-bound Pf-MATE (an archaeal DinF protein) also does not reveal an acidic cluster surrounding the Br-NRF ligand.

(page 7, paragraph 1) Based on structural comparisons to Pf-MATE and NorM-VC, the authors state that the bend in TM7 is the most distinct point from other reported MATE structures. However, Figure 1b shows that TM7 in NorM-VC is also not straight (i.e. bent). Moreover, the occluded nature of the extracellular access cavity in AtDTX relative to those present in both Pf-MATE and NorM-VC may suggest that the crystallization conditions may have induced a conformation that is distinct from Pf-MATE and NorM-VC. Could this be true?

(page 8, paragraph 2) To probe the importance of an identified H-bonding network, the authors generated mutations in hMATE based on the AtDTX structure (and sequence similarity) and performed functional analyses based on TEA transport. Decreased activities for all of the mutants lead the authors to believe that the H-bonding network is important. The paper would be strengthened if the mutations were generated in AtDTX instead of (or in addition to) the hMATE protein. This might be important to validate the structural extrapolation between the two systems. Also, it's not clear what role is being ascribed to the H-bonding network. To simply say that it's important for transport, in the reviewer's opinion, can mean many things or nothing. Since the functional data are used to formulate a mechanistic model, the data should better reveal structural or functional roles. Does the H-bonding network play a functional or structural role? The fact that each position was probed with a single substitution makes the authors' interpretations less compelling. Site-directed protein modifications that conserve certain properties while varying others could in principle reveal more specific information regarding the roles of particular residues. Also, EVERY SINGLE substitution resulted in decreased transport; control substitutions are necessary to validate the reported observations. Substrate-binding studies might also be of use.

The above comments apply to the mutational results that probe the flexibility of TM1 (page 9, paragraph 2), the Trp residue that occludes the H-bonding network from the internal cavity (page 8, paragraph 3) and substrate access from the intercellular and extracellular gates (page 10, paragraph 2). With respect to the latter, the paper presenting the MurJ structure described a very nice method that identifies substrate/solvent access portals in the MATE proteins.

To investigate the inward-facing conformation of AtDTX14, the authors generate a model based on the MurJ structure, which importantly, contains 14 TMS, not 12, as observed with the other MATE proteins. This importance difference is not noted. The authors do not describe how the (homology - ??) model was generated. However, the authors do state that the model was verified using the program EV-fold_membrane (Figures 4e and 4f and Figure S9). It is not clear how the EC_coupling values presented speak to the validity of the model. The authors do not explain this. Biochemical validation of the model is certainly lacking. The paper describing the MurJ structure discusses a possible method that might better address the model. The authors generate and test a single mutant G285Q to interrogate the model. Decreased activity of xenobiotic transport by AtDTX14 is enough to convince the authors to assign a specific functional role to that residue. It is not clear that the Gly to Gln substitution is appropriate nor is it clear that a functional role can be

assigned based on such little data, especially a loss of function. The authors might use residue side-chains of varying sizes (or ligands of different sizes) to observe variable losses of transport. The authors also do not discuss other important, relevant consequences of the interconversion between the outward- and inward-facing AtDTX conformations (i.e. what happens near the substrate-binding pocket?) This information might be important for supporting the proposed mechanism described on page 13 on the manuscript.

Figure 2. Hydrogen-bonding distances should be indicated. Also, some of the proposed H-bonding contacts do not appear to be possible due to angular requirements.

The AtDTX14 structure was solved by molecular replacement. How did the authors obtain an unbiased electron map (shown in Supplementary Figure 3.) This should be clarified.

Specific comments by Referee #1:

This manuscript presents a high resolution structure of the first eukaryotic MATE (eMATE) transporter, AtDTX14 from Arabidopsis thaliana. As for most transporters of the MATE family, well diffracting crystals were obtained in cubo. In fact, a mutation of a highly conserved proline in transmembrane helix 1 (TM1) into alanine was required to obtain the crystals. The structure is compared to various other outward-facing bacterial MATE transporters, belonging to the DinF and NorM subfamily, respectively. The major finding of the study is the presence of a conserved hydrogen bonding network within the C-lobe of AtDTX14, shielded by a Trp-residue, which is reminiscent of a similar network found in the C-lobe of the NorM subfamily and the symmetry-related N-lobe of the DinF family. In members of the DinF family, a corresponding hydrogen bonding network in the N-lobe had been postulated to be responsible for bending movements in TM1 as a consequence of protonation/deprotonation of conserved acidic residues and thereby driving substrate transport. By comparing structures of AtDTX14 with the NorM subfamily member NorM-VC, an analogous bending movement in the symmetry related TM7 was observed. While transport of novobiocin by AtDTX14 was demonstrated in E. coli, the transport activity appeared too low for conducting functional assays with AtDTX14. Therefore, the authors resorted to human hMATE1 for their functional studies, for which a robust in vivo transport assay of tetraethylammonium (TEA) exists and interrogated residues of the hydrogen bonding network of the C-lobe as well as glycine residues places at the intra- and extracellular gates closed by the two lobes when the transporter adopts its inward- as well as its outward-facing state.

As the Referee pointed out, the norfloxacin transport activity of AtDTX14 is actually weaker than that of AcrB, which is a positive control in our experiment. However, the activity of AtDTX14 is robust enough to perform the structure-guided mutational analysis. Thus, in the revised manuscript, we added the results of the mutational analysis of AtDTX14, which strongly supported our discussion (Fig. 1a, b).

From a structural biology point of view and given the high resolution obtained, the work is excellent and provides novel insights into a new subfamily of MATE transporters, the eMATEs. From a functional point of view, the authors provide some evidence for their hypotheses. However, these are solely based on a single type of in vivo transport assays conducted with a homologous transporter. While the quality of the figures is excellent, the manuscript is somewhat more difficult to digest, presumably due to the quite lengthy structural descriptions, which are not that attractive to read.

Nevertheless, the manuscript is clearly written.

As described above, we newly added the functional analysis of AtDTX14 to provide more robust evidence for our hypothesis. Furthermore, we reorganized and rewrote the manuscript to clearly explain our results and discussion. We believe that the revised manuscript is easier to digest and more attractive to read.

Major points

1) The authors compare the AtDTX14 structure with NorM-VC and draw conclusions regarding the functional importance of TM7 bending. Therefore it seems that eMATEs are more closely related to the NorM subfamily of MATE transporters as opposed to the DinF subfamily. A phylogenetic tree including members of the three MATE subfamilies would be very helpful to clarify this question. Should the eMATEs not be that closely related to the NorM subfamily, I consider it as not entirely adequate to draw mechanistic conclusion from a comparison between structures of AtDTX14 and NorM-VC.

The eMATE and NorM subfamilies share the functionally important acidic residues in the C-lobe, while the DinF subfamily members lack them. This point (*i.e.*, possible functional similarity between eMATE and NorM) has also been discussed previously (Radchenko *et al.*, *Nat. Commun.*, 6, 7995 (2015)). Therefore, our hypothesis is mainly based on the structural comparison between eMATE and NorM members. In the revised manuscript, we clarified this point in the Introduction (Line 91-93), and in the Results and Discussion sections (Line 184-208). We also revised the supplementary figure of the a. a. sequence alignment (Suppl. Fig. 1), to clearly show the similarity between eMATE and NorM.

2) All MATEs have so far been crystallized in outward-facing states with a closed intracellular gate. Recently a structure inward-facing MurJ (belonging as the MATEs to the MOP superfamily of transporters) was described, based on which the authors made a homology model. At both the intra- and extracellular gate the authors spotted conserved glycine residues which were mutated into bulky glutamine residues in hMATE1. The corresponding mutants were almost inactive in terms of transport. However, replacing a conserved glycine by a bulky glutamine likely results in major structural disturbances. The authors at least have to show whether the respective mutants (introduced in AtDTX14) can still be purified and behave the same as the wildtype protein on size exclusion chromatography analysis.

In these Gly-to-Gln mutations, we intended to introduce a bulky side chain to the corresponding sites; however, we cannot rule out the possibility that this mutation introduced large structural disturbances by also changing the main-chain conformations, as the Referee pointed out. Therefore, to clarify the main point of our manuscript, we removed the results of the Gly-to-Gln mutants from the revised manuscript, as they are not directly relevant to the main point of our manuscript, the substrate transport mechanism by the structural change in the C-lobe and the conserved acidic residues.

3) *The inward-facing homology model was validated by calculating the evolutionary coupling pairs. However, it should in addition be validated by strategically placed cysteine pairs, which only can form a disulfide bond when the transporter adopts its presumed inward-facing state.*

Prior to the Cys crosslink experiment, we have to mutate all of the native Cys residues (73, 92, 119, 188, 194, 233, 263, 292, 441 in AtDTX14). To search for the combinations of mutations that do not affect the expression and mono-dispersity of the protein, we have to try many combinations of mutations in each site. Furthermore, since the AtDTX14 protein cannot be expressed in *E. coli*, we have to prepare all of these trial mutants using the HEK293 or insect cell expression system. Thus, the Cys crosslinking analysis of AtDTX14 is not only time-consuming, but also a difficult and challenging experiment. In contrast, the inward-open model structure of AtDTX14 is based on the inward-open crystal structure of the known homologous transporter, which is sufficient to support the main topic in our manuscript. Therefore, we think that it is beyond the scope of the present manuscript to perform the technically difficult Cys crosslinking experiment just to support the inward-open model. Instead, we shortened the description about the outward gate formation, and focused more on the comparison to the bacterial MATE transporters in the revised manuscript.

4) *Supplementary Figure 2: The novobiocin resistance conferred by AtDTX14 in E. coli seems to be very weak. In addition, the picture is only partially self-explanatory and difficult to understand even with the legend. I had difficulties to understand that the numbers provided in the right panel correspond to the constructs shown on the left panel. Further, I do not understand what can be learned from the left panel. The strains contain the respective constructs on a plasmid and grow fine in the absence of inducer IPTG and novobiocin, i.e. I do not see any insight one can gain from this. Finally, I would see it as mandatory to introduce mutations into AtDTX14, which abrogate its function (e.g. E265Q or D383N) to validate the results obtained in the presence of novobiocin +*

IPTG as shown on the right panel at the bottom.

As described above, the activity of AtDTX14 is sufficiently robust to perform the structure-guided mutational analysis. In the revised manuscript, we added the results of the mutational analysis of AtDTX14, including the mutant of the conserved acidic residue (D383A), which now strongly support our discussion (Fig. 1). In addition, we performed the fluorescence-based SEC analysis of these mutants, to confirm their structural integrity (Supplementary Fig. 6).

5) Concerning the interaction between E265 and D383: this may in fact represent a low-barrier hydrogen bond. According to figure 5, the distance is 2.69 Å, which rather points towards a classical hydrogen bond. This interaction may deserve a bit of deeper thinking.

The hydrogen bonding distance of 2.69 Å seems to be a weak rationale to conclude that these two residues form a low-barrier hydrogen bond. The structure refinement statistics showed that the coordinate error of the present crystal structure is 0.37 Å. In a rough estimate, the deviation of this distance from the standard O-H:O hydrogen bond (~2.8 Å) is about 0.1 Å, which does not seem to be significant, given this coordinate error. Indeed, there are several main-chain (N-H:O=C) hydrogen bonds with distances around 2.7 Å in the present crystal structure.

With respect to the model in Figure 6, it is important to indicate which state corresponds to the crystal structure (outward-open “bent”). Further, it would be nice to have a depiction of the conserved Trp residue, which shields the collapsed C-lobe cavity in the outward-open “bent” structure.

We indicated the state of the present crystal structure in Fig. 8, including the conserved Trp residue.

Minor points

1) Concerning the mutations in the hydrogen bonding network, in particular E265 and D383 (AtDTX14 numbering), it is interesting that the authors did exchange the residue into Gln and Asn, respectively. This was a very good choice, because in this way the mutation only removed the protonatable group without changing the side chain length. Perhaps the authors should mention that in the previous publication they cite (Ref 25), the respective residues were exchanged for aspartates and alanines.

We mentioned the Glu to Gln (and Asp to Asn) mutations of the conserved acidic residues (Line 235-250).

2) Concerning the pK of the acidic residues within the C-lobe, it may be worth mentioning that in a hydrophobic environment, the aspartates and glutamates may in fact exhibit pKa in the range of 6-7. This has been shown for many proton translocating transporters including the F1Fo-ATPase and RND pumps such as AcrB. By removing the conserved Trp residues, the pKa values of E265 and D383 may be much decreased, thereby explaining the loss of function.

We included a discussion about the pKa of the acidic residues and the hydrophobic environment in the revised manuscript (Line 218).

Specific comments by Referee #2:

The manuscript, "Structural basis for xenobiotic extrusion by eukaryotic MATE transporters" reports the X-ray structure of the eukaryotic MATE transporter, AtDTX14 (AtDTX), from Arabidopsis thaliana at 2.6Å;. The authors employ structure-based comparisons of AtDTX with MATE proteins belonging to other subfamilies, including NorM, DinF and MurJ to guide structure-function studies of a human homolog hMATE, identify potentially important functional-structural determinants and ultimately propose possible a mechanism for xenobiotic recognition and transport by AtDTX.

Importantly, this appears to be the first X-ray structure of a protein from the eMATE family. Structural departures from the bacterial MATE proteins and their mechanistic consequences could also be important if understood at the molecular level. The manuscript aims to achieve such a description of both structure and mechanism.

Unfortunately, the manuscript does not at all read very clearly nor does it follow a clear logical narrative. Many of the details of the AtDTX structure are lost in the very detailed (sometimes--maybe inappropriate) comparisons with MATE proteins belonging to other subfamilies.

We reorganized and rewrote the manuscript to clearly explain our results and discussion. We also described the rationale of the structural comparison with the other bacterial MATE transporters. We believe that the revised manuscript is now clear and logical.

The study is also weakened by the fact that mutational studies are not used to probe the reported

AtDTX structure. Instead, they are used to probe a homologous protein, hMATE. The reviewer feels that the minimal set of reported mutational data with the single AtDTX structure are not sufficient to formulate the mechanism proposed for transport (see discussion below). Additional structures, including a substrate-bound AtDTX as well as mutational studies on the plant system might afford a more compelling manuscript.

We newly added the functional analysis of AtDTX14 to provide more robust evidence (Fig. 1). The results of the analysis now strongly support our hypothesis in the manuscript.

Finally, the mechanistic model proposed for xenobiotic transport is very similar to that proposed for DinF (Lu (2013) NSB 20, 1310). The DinF transport mechanism also includes structural changes in TM7 (see manuscript abstract). As such, the novelty of the results and mechanistic proposal are questionable.

In the *NSMB* paper by Lu *et al.*, both TM7 and TM8 are bent and the extracellular half of the TM7-TM8 hairpin forms contacts with the N-lobe (Fig. 2 in *Nat. Struct. Mol. Biol.* 20, 1310–7 (2013)). As a result, the extracellular half of the C-lobe is split into two parts, which were hypothesized to be important for the transport mechanism of DinF-BH. They discussed that this structural change is just a step in the transition between the occluded and outward-open states, and the structural changes in TM7 are not coupled to substrate/co-substrate binding. In contrast, in our paper, we proposed that only TM7 is bent towards the TM8 side (Fig. 2c), thereby collapsing the substrate binding site formed between TM7 and TM8 (Fig. 7, Supplementary Fig. 10) to enable the substrate extrusion to the outside of the cell. Thus, in the eMATE mechanism, TM8 is not bent, and TM7 and TM8 do not contact the N-lobe. These points are substantially different from the previous hypothesis for DinF-BH proposed by Lu *et al.* Furthermore, our paper discusses that this structural change in TM7 is coupled to the protonation of the transporter, thus explaining the proton-coupled antiport mechanism by eMATE. This important point is completely different from the mechanism of DinF-BH proposed by Lu *et al.* Thus, the previously proposed mechanism of the bacterial DinF transporter does not compromise the novelty of our findings. To clarify this point, we added a figure and discussion comparing the structural differences between eMATE and DinF-BH (Supplementary Fig. 12).

(page 4, paragraph 2) The authors claimed to have identified the cation binding pocket in AtDTX based on the (1) presence of two acidic residues (called a conserved acidic cluster) and (2) binding of Rb⁺ and Cs⁺ ions to NorM in a cavity lined by two homologous acidic residues. However, ligand-bound structures of NorM do not show lipophilic cations bound to the same region as the

Rb⁺ and Cs⁺ ions (pdb ids: 4huk (P4P), 4hum (ET), 4hun (RHQ), 4hul (Cs), 3mku (Rb) (NorM) as well as 4lz6, 4lz9 (RHQ), 3vvp (Br-NRF) (DinF).

In general, the substrate binding site of the transporters with the alternate accessing mechanism is conserved in all of the inward-open, occluded and outward-open conformations. Its accessibility to the intra/extracellular spaces is controlled by the opening/closing motions of the intra/extracellular gates, which enable the coupled transport of the substrate and co-substrate (*Annu. Rev. Biochem.* 85, 1, 543-572 (2016), *Nat. Rev. Microbiol.* 12, 79–87 (2013)). Thus, the substrate binding site is usually situated around the middle of the TM segments.

In contrast to this general mechanism, the organic cation substrates of NorM-NG reported by Lu *et al.* (*PNAS* 110, 2099-2014, 2013; PDB IDs: 4HUK, 4HUM, 4HUN) are bound to the site between the extracellular half of the N- and C-lobes, where the extracellular gate is formed in the inward-open MurJ structure and our inward-open model of eMATE (Suppl. Fig. 11). If this NorM-NG site reported by Lu *et al.* were the substrate binding site, then the inward-open structure would be impossible, because the N- and C-lobes would clash with the bound substrate. Instead, it is quite likely that the substrates are not bound to the substrate binding site, but are transiently trapped in a minor site in the transport pathway from the substrate binding site to the extracellular space, in these NorM-NG structures.

Therefore, in our manuscript, we proposed the hypothesis that the cavity between TM7, TM8 and TM10 in the C-lobe is a substrate binding pocket, which is different from the site where the substrate is observed in the NorM-NG crystal structure. This substrate binding site is conserved in both the inward- and outward-open conformations, and substrate binding to this site does not hamper the structural conversion between the inward- and outward-open conformations (Supplementary Fig. 10). Moreover, this site is also consistent with previous functional analyses of hMATE1 (*Am. J. Physiol. Cell Physiol.* 294, C1074-8 (2008)) and NorM-PS (*J. Biol. Chem.* 291, 15503–15514 (2016)).

The above discussion about the substrate binding pocket is missing in the original manuscript, so we added a discussion about the comparison to the previous studies in the revised manuscript, to clearly explain this point.

Also, an analysis of the RH6G-bound DinF structure does not show an acidic cluster near RH6G. Moreover, a Br-NRF-bound Pf-MATE (an archaeal DinF protein) also does not reveal an acidic cluster surrounding the Br-NRF ligand.

In our manuscript, we proposed that eMATE has a different substrate binding site and extrusion mechanism from those of DinF, based on the structure and sequence comparisons of the

reported MATE structures. Thus, it is not surprising that the substrate binding site of eMATE has distinct properties from those of the DinF subfamily transporters (*i.e.*, DinF-BH and PfMATE).

(page 7, paragraph 1) Based on structural comparisons to Pf-MATE and NorM-VC, the authors state that the bend in TM7 is the most distinct point from other reported MATE structures. However, Figure 1b shows that TM7 in NorM-VC is also not straight (i.e. bent).

As mentioned above, the main point of our paper is that TM7 is bent toward the TM8 side (Fig. 2c), thereby collapsing the substrate binding cavity formed between TM7, TM8, and TM10 (Fig. 7 and Supplementary Fig. 10). Indeed, the conformation of TM7 in NorM-VC slightly deviates from that of an ideal α helix, as Referee #2 pointed out. However, the large substrate binding cavity is still formed between TM7 and TM8 in the C-lobe of NorM-VC (Fig. 2b and Fig. 7b). This slight deviation does NOT collapse the substrate binding cavity of NorM-VC. Therefore, this slight deviation of TM7 in NorM-VC from an ideal α helix neither affects the transport model proposed in our paper nor compromises the novelty of our findings. To clearly visualize this difference, we revised the figures, and included the quantitative comparison of the bending angles of the TM7 helices (intra- and extracellular halves) between AtDTX14 and NorM-VC (Fig. 2c).

Moreover, the occluded nature of the extracellular access cavity in AtDTX relative to those present in both Pf-MATE and NorM-VC may suggest that the crystallization conditions may have induced a conformation that is distinct from Pf-MATE and NorM-VC. Could this be true?

One possible explanation for this difference may be the difference in the crystallization conditions, as the Referee pointed out. However, AtDTX14, PfMATE, and NorM-VC belong to a different subfamily and are from different species with different amino-acid sequences. Thus, it is also likely that their energy landscapes and ground states may be different, which resulted in the different conformations in the crystals. In any case, it is possible to say that the present eMATE structure represents the partially-occluded outward-open state.

(page 8, paragraph 2) To probe the importance of an identified H-bonding network, the authors generated mutations in hMATE based on the AtDTX structure (and sequence similarity) and performed functional analyses based on TEA transport. Decreased activities for all of the mutants lead the authors to believe that the H-bonding network is important. The paper would be strengthened if the mutations were generated in AtDTX instead of (or in addition to) the hMATE protein. This might be important to validate the structural extrapolation between the two systems.

As described in the response to Referee #1, we newly added the functional analysis of AtDTX14 to provide more robust evidence for our hypothesis.

Also, it's not clear what role is being ascribed to the H-bonding network. To simply say that it's important for transport, in the reviewer's opinion, can mean many things or nothing.

As discussed in the Discussion section in the main text, the H-bonding network involving the conserved acidic residues enables the structural change of TM7 upon proton binding, which explains the proton-coupled substrate extrusion mechanism of eMATE. As the Referee pointed out, we certainly described the results of the mutational analysis as “important for transport” in the Results section (Line 201-208), since the mutational results indicated the importance of the target residues in the transporter function. However, by comprehensively integrating the crystal structure and the results of the functional analyses of this paper and previous work, we can draw the overall picture of the transport mechanism, which is described in the Discussion section in our manuscript (Line 296-318).

Since the functional data are used to formulate a mechanistic model, the data should better reveal structural or functional roles. Does the H-bonding network play a functional or structural role? The fact that the each position was probed with a single substitution makes the authors' interpretations less compelling. Site-directed protein modifications that conserve certain properties while varying others could in principle reveal more specific information regarding the roles of particular residues. Also, EVERY SINGLE substitution resulted in decreased transport; control substitutions are necessary to validate the reported observations. Substrate-binding studies might also be of use.

In the present manuscript, we not only measured the transport activities of the hMATE1 mutants, but also analyzed their expression and membrane localization (Suppl. Figs. 8, 9). For example, the E273Q mutant does conserve certain properties (expression and membrane localization, *i.e.*, structural stability), while varying others (transport activity), thereby revealing specific information regarding its role; and E273 (E265 of AtDTX14) in the H-bonding network has a functional role, *i.e.*, the proton binding site, rather than a structural role.

Furthermore, as you can see in Figure 5b, c, NOT every single substitution resulted in decreased transport. For example, the Y416A (Y410 in AtDTX14) mutation had a moderate effect as compared to the other mutations, indicating that this Tyr residue does not play a critical role in the transport mechanism. To clarify this point, we added a description about the mutations with a moderate effect (Line 243-244).

The above comments apply to the mutational results that probe the flexibility of TM1 (page 9, paragraph 2), the Trp residue that occludes the H-bonding network from the internal cavity (page 8,

paragraph 3) and substrate access from the intercellular and extracellular gates (page 10, paragraph 2).

As described above, we investigated the properties of the mutants not only by the transport activities, but also by their expression and membrane localization. Furthermore, we also tested the transport activities of both AtDTX14 and hMATE1, using two different assay systems. We believe that these multiple approaches are enough to support our hypothesis of the transport mechanism.

With respect to the latter, the paper presenting the MurJ structure described a very nice method that identifies substrate/solvent access portals in the MATE proteins.

As the referee pointed out, the experiment using a Cys-modifying reagent (MTSES in the case of MOP) would be useful to examine the substrate/solvent accessibility. However, prior to the Cys modification experiment, we have to mutate all of the native Cys residues (73, 92, 119, 188, 194, 233, 263, 292, 441 in AtDTX14). To search for the combinations of mutations that do not affect the expression and mono-dispersity of the protein, we have to try many combinations of mutations in each site. Furthermore, since the AtDTX14 protein cannot be expressed in *E. coli*, we have to prepare all of these trial mutants using the HEK293 or insect cell expression system (also note that MOP is a bacterial protein which can be prepared *E. coli* expression system). Thus, the Cys-modification analysis of AtDTX14 is not only time-consuming, but also a difficult and challenging experiment. Therefore, we think that it is beyond the scope of the present manuscript to perform the technically difficult Cys modification experiment just to support the substrate accessibility, which is evident from the crystal structure.

To investigate the inward-facing conformation of AtDTX14, the authors generate a model based on the MurJ structure, which importantly, contains 14 TMS, not 12, as observed with the other MATE proteins. This importance difference is not noted.

In the MurJ structure report (*Nat. Struct. Mol. Biol.* 24, 171–176 (2017)), the additional TM13 and TM14 are suggested to be important for capturing the substrate lipid acyl chain; however, the substrate of eMATE is not a lipid. Indeed, the Introduction section of this paper stated that “The canonical MOP transporter core consisting of 12 transmembrane helices (TMs), MurJ has two additional C-terminal TMs (13 and 14) of unknown function.” Therefore, these TM13 and TM14 of MurJ are involved in the MurJ-specific function, and have no relation to the common function of MOP superfamily transporters. It is obvious that the function of TM13 and TM14 and the difference between 12TM vs. 14TM are totally unrelated points in creating the inward-facing model of eMATE, based on the MurJ crystal structure. Therefore, this difference is not discussed in our manuscript.

The authors do not describe how the (homology-??) model was generated.

We included the description about how we created the inward-open model structure in the revised manuscript (Line 166-167).

However, the authors do state that the model was verified using the program EV-fold_membrane (Figures 4e and 4f and Figure S9). It is not clear how the EC_coupling values presented speak to the validity of the model. The authors do not explain this.

We added a short description about how the EC values support the validity of the model structure of the inward-open form (Line 176-179). However, for further information about the details of the EV_fold and the EC value, readers should refer to the literature cited in the main text.

Biochemical validation of the model is certainly lacking. The paper describing the MurJ structure discusses a possible method that might better address the model.

To validate the inward-open model, a Cys crosslinking experiment is appropriate. However, prior to the Cys crosslinking experiment, we have to mutate all of the native Cys residues (73, 92, 119, 188, 194, 233, 263, 292, 441 in AtDTX14). To search for the combinations of mutations that do not affect the expression and mono-dispersity of the protein, we have to try many combinations of mutations in each site. Furthermore, since the AtDTX14 protein cannot be expressed in *E. coli*, we have to prepare all of these trial mutants using the HEK293 or insect cell expression system. Thus, the Cys crosslinking analysis of AtDTX14 is not only time-consuming, but also a difficult and challenging experiment. In contrast, the inward-open model structure of AtDTX14 is based on the inward-open crystal structure of the known homologous transporter, which is sufficient to support the main topic in our manuscript. Therefore, we think that it is beyond the scope of the present manuscript to perform the technically difficult Cys crosslinking experiment just to support the inward-open model. Instead, we shortened the description about the outward gate formation, and focused more on the comparison to the bacterial MATE transporters in the revised manuscript.

The authors generate and test a single mutant G285Q to interrogate the model. Decreased activity of xenobiotic transport by AtDTX14 is enough to convince the authors to assign a specific functional role to that residue. It is not clear that the Gly to Gln substitution is appropriate nor is it clear that a functional role can be assigned based on such little data, especially a loss of function. The authors might use residue side-chains of varying sizes (or ligands of different sizes) to observe variable losses of transport.

In these Gly-to-Gln mutations, we intended to introduce a bulky side chain to the corresponding sites; however, we cannot rule out the possibility that this mutation introduced large

structural disturbances by also changing the main-chain conformations. Therefore, we removed the results of the Gly-to-Gln mutants from the revised manuscript, to clarify the main point of our manuscript (*i.e.*, the substrate transport mechanism).

The authors also do not discuss other important, relevant consequences of the interconversion between the outward- and inward-facing AtDTX conformations (i.e. what happens near the substrate-binding pocket?) This information might be important for supporting the proposed mechanism described on page 13 on the manuscript.

We included the figures showing that the proposed substrate-binding site is conserved in both the outward- and inward-facing structures in the revised manuscript (Supplementary Fig. 10).

Figure 2. Hydrogen-bonding distances should be indicated. Also, some of the proposed H-bonding contacts do not appear to be possible due to angular requirements.

We corrected the figure as the Referee requested (Fig. 4).

The AtDTX14 structure was solved by molecular replacement. How did the authors obtain an unbiased electron map (shown in Supplementary Figure 3.) This should be clarified.

The present crystal structure is free from the bias of the model structure (NorM-VC) used in the molecular replacement, as indicated by the statistics of the structure refinement (*e.g.*, $R_{\text{work}}/R_{\text{free}}$), the resolution, and the quality of the electron density map. The word, “unbiased”, in the Supplementary Figure 2 legend means that the bias originating from the structure factor (F_c) of the present AtDTX14 model (and not from the model used in the molecular replacement) is alleviated by introducing the model-incompleteness coefficient D and the figure-of-merit m , as compared to the legacy $2F_o - F_c$ map (see Bernhard Rupp, *BIOMOLECULAR CRYSTALLOGRAPHY*, p.618-619).

Reviewers' comments:

Reviewer #1 (Remarks to the Author):

The revised version of the manuscript is clearly improved and the authors have invested major efforts in conducting norfloxacin resistance experiments with AtDTX14 (new Figure 1) and in clarifying both reviewers' questions regarding the mechanism of transport. The article is now really well written and contains very nice figures. The proposed transport mechanism (now also featuring the conserved Trp residue) is feasible and supported by the data.

I have two minor final points to be addressed:

1) On line 319 ff the authors write:

To examine the effect of the
320 structural conversion from the outward- to inward-open states on the C-lobe cavity, we
321 created the inward-open model structure with the straight TM7 conformation. The
322 model showed that the C-lobe cavity is also conserved in the inward-open model
323 structure (Supplementary Fig. 10a, b), and its accessibility to the intra- and
324 extracellular spaces is controlled by the rocking motion of the N- and C-lobes, which
325 opens and closes the extra- and intracellular gates.

I assume the model was constructed based on the coordinates of MurJ (not mentioned). It is a bit confusing to call the C-lobe cavity as being conserved in this inward-open model. One may say that the C-lobe cavity is present in MurJ and hence also in the inward-facing model. Should it be the case that the model was not constructed based on MurJ but by other means, this needs to be specified more clearly. The material and methods section does not contain information on the construction of the inward-facing homology model of AtDTX14.

2) Line 208: These conserved acidic residues may function as the proton binding site for the driving force of eMATE.

This sounds a bit unprecise. My suggestion would be:

Proton binding at these conserved acidic residues is likely to play a key role in coupling the proton-motive force to substrate extrusion in eMATE.

Reviewer #2 (Remarks to the Author):

Remarks to the Author:

Regarding the revisions to the manuscript "Structural basis for xenobiotic extrusion by eukaryotic transporter," I do not feel that comments were adequately addressed the concerns of the reviewer. In some cases, the authors were almost dismissive in their responses.

Reviewer 1 asked for the authors to provide a phylogenetic tree that included the proteins from the different MATE families to justify comparisons made in the text. Such a tree was not provided and the generation of such would have taken very little effort. Moreover, such a tree would have been informative.

Both reviewers asked for biochemical validation of the inward facing homology model. The authors

provide a long, drawn out explanation of why such an experiment would not be “worth the trouble.” Specifically, the authors state,

“...Thus, the Cys crosslinking analysis of AtDTX14 is not only time-consuming but also a difficult and challenging experiment. In contrast, the inward-open model structure of AtDTX14 is based on the inward-open crystal structure of the known homologous transporter, which is sufficient to support the main topic in our manuscript...”

I disagree with the author’s opinion that the generation model of based on a known homologous transporter (MurJ) is both necessary and sufficient to support the main topic of the manuscript. Moreover, the actual degree of homology between the transporters is never stated. Also, the structural divergence between the two proteins (MurJ has 14 TMHs whereas both hMATE and AtDTX14 have 12) raises questions regarding the accuracy throughout the entire structure (homology models often contain regions that have higher degrees of uncertainty than others). Also, the EC coupling values do appear to be consistent with the location of the residues defining the inward and outward gates; however, they do not validate the remaining regions of the modeled protein, which also appear to be important to their proposed mechanism.

To the reviewer’s comment, “The authors do not describe how the (homology-??) model was generated,” the authors gave the following hardly acceptable response,

We included the description about how we created the inward-open model structure in the revised manuscript (Line 166-167).

“To investigate the inward-open conformation of the eukaryotic MATE, we generated the inward-open model structure of AtDTX14, by separately superimposing the N- and C-lobes of AtDTX14 onto the MurJ structure (Fig. 3b, e, f).”

Without the aid of structural data (that of a ligand-bound AtDTX14), the authors appear to sometimes over-interpret mutational results

For example, “In contrast, the mutation of Gln449 (Gln443) affected both the TEA and cimetidine transport activities, and the mutation of Asn412 (Asn406) strongly reduced the TEA transport activity (Fig. 5b, c). These results suggested that Asn412 is directly involved in the recognition of TEA and TEA-like substrates (Fig. 5b, c).”

The observed effects could result from both indirect or direct effects wherein specified residues play functional or structural roles. The authors include gel-filtration data to support structural effects. However, whereas the inclusion of such show that the describe variants do indeed fold, it does not distinguish between the functional or structural effects.

For the proposed mechanism, the authors state that “...the protonation of acidic residues causes an electrostatic attraction between them...” This appears to be a critical feature of the proposed mechanism. This statement does not describe a likely scenario, making the reviewer skeptical of the proposed transport mechanism.

The authors state that hydrogen-bonding distances were added to Figure 4. THEY WERE NOT!

In response to the reviewer’s comment, “The authors claimed to have identified the cation binding pocket in AtDTX based on the (1) presence of two acidic residues (called a conserved acidic cluster) and (2) binding of Rb⁺ and Cs⁺ ions to NorM in a cavity lined by two homologous acidic residues. However, ligand-bound structures of NorM do not show lipophilic cations bound to the same region as the Rb⁺ and Cs⁺ ions (pdb ids: 4huk (P4P), 4hum (ET), 4hun (RHQ), 4hul (Cs), 3mku (Rb) (NorM) as well as 4lz6, 4lz9 (RHQ), 3vvp (Br-NRF) (DinF),” the author’s simply state that the previously ligand-bound structures incorrectly identify the substrate-binding sites.

Without a ligand-bound structure of their own or without concrete proof of that statement, such a case cannot be made. Moreover, the authors provide no compelling evidence (modeling or otherwise) that ligands indeed bind to their proposed site.

To the reviewer's comment, "The AtDTX14 structure was solved by molecular replacement. How did the authors obtain an unbiased electron map (shown in Supplementary Figure 3.) This should be clarified," the authors responded,

"The present crystal structure is free from the bias of the model structure (NorM-VC) used in the molecular replacement, as indicated by the statistics of the structure refinement (e.g., R_{work}/R_{free}), the resolution, and the quality of the electron density map. The word, "unbiased", in the Supplementary Figure 2 legend means that the bias originating from the structure factor (F_c) of the present AtDTX14 model (and not from the model used in the molecular replacement) is alleviated by introducing the model-incompleteness coefficient D and the figure-of-merit m , as compared to the legacy $2F_o-F_c$ map (see Bernhard Rupp, BIOMOLECULAR CRYSTALLOGRAPHY, p.618-619)."

The map that is shown by the authors is not completely free from bias and is commonly called a "cross-validated sigma-A weighted map." Maps generated with the coefficients $2mF_o-DF_c$ are not "unbiased". Composite omit maps, simulated annealing omit maps and maps generated via de novo phasing are unbiased. Maps from which the phases are generated via the phases from another structure (via molecular replacement) are biased. Structure refinement (structure modeling) itself introduces bias. Structure refinement statistics do not necessarily indicate bias, they can often reveal problems due to over-fitting, which includes many types of errors. As such, the map shown should not be called an unbiased map. The reviewer was not stating that the map was inaccurate or invalid, only that the word unbiased should not be used.

Overall, the paper presents a structure of an eMATE transporter that sequence divergences from of known MATE transporters. The sequence divergences result in novel features that may or may not play direct roles in transport. Whereas the reviewer believes that authors validate the observed features, their actual functional roles are not revealed clearly elucidated by the limited amount of mutagenesis data presented. Without additional structures, including one presenting the ligand-bound form of AtDTX14, their proposed mechanism is not compelling. This is especially true considering their description of the protonation-driven helix conformational changes.

Furthermore, the authors might consider a mechanism wherein proton-coupled export proceeds through a state in which both protons and drugs are bound. Allosteric coupling usually requires that the binding of protons (downhill) effects the binding of drugs (uphill). The reviewer is aware that mechanisms similar to the one proposed here have been previously published. However, the publication of a mechanism does not establish its validity. Thermodynamic intuition (at least, reviewers) suggest that the proposed mechanism cannot be correct (especially in light of the authors' description of the proton-related effects on protein structure).

Specific comments by Referee #1:

1) On line 319 ff the authors write:

To examine the effect of the structural conversion from the outward- to inward-open states on the C-lobe cavity, we created the inward-open model structure with the straight TM7 conformation. The model showed that the C-lobe cavity is also conserved in the inward-open model structure (Supplementary Fig. 10a, b), and its accessibility to the intra- and extracellular spaces is controlled by the rocking motion of the N- and C-lobes, which opens and closes the extra- and intracellular gates.

I assume the model was constructed based on the coordinates of MurJ (not mentioned). It is a bit confusing to call the C-lobe cavity as being conserved in this inward-open model. One may say that the C-lobe cavity is present in MurJ and hence also in the inward-facing model. Should it be the case that the model was not constructed based on MurJ but by other means, this needs to be specified more clearly. The material and methods section does not contain information on the construction of the inward-facing homology model of AtDTX14.

Thank you very much for the favorable comments on our manuscript. According to the Referee's suggestion, we described how the model structures were generated in more detail in the legends of Fig. 3 and Supplementary Fig. 10 (p.31, 1.3–8; p.38, 1.18–p.39, 1.3).

2) Line 208: *These conserved acidic residues may function as the proton binding site for the driving force of eMATE.*

This sounds a bit unprecise. My suggestion would be:

Proton binding at these conserved acidic residues is likely to play a key role in coupling the proton-motive force to substrate extrusion in eMATE.

We corrected the manuscript as the referee suggested (p.10, 1.13–15)

Specific comments by Referee #2:

Regarding the revisions to the manuscript "Structural basis for xenobiotic extrusion by eukaryotic transporter," I do not feel that comments were adequately addressed the concerns of the reviewer. In

some cases, the authors were almost dismissive in their responses.

Reviewer 1 asked for the authors to provide a phylogenetic tree that included the proteins from the different MATE families to justify comparisons made in the text. Such a tree was not provided and the generation of such would have taken very little effort. Moreover, such a tree would have been informative.

Thank you very much for your constructive comments on our manuscript. According to the Referee's suggestion, we included the phylogenetic tree diagram in Supplementary Fig. 1b. Now the tree diagram clearly indicates that the MATE family is divided into the three subfamilies.

Both reviewers asked for biochemical validation of the inward facing homology model. The authors provide a long, drawn out explanation of why such an experiment would not be "worth the trouble." Specifically, the authors state,

"...Thus, the Cys crosslinking analysis of AtDTX14 is not only time-consuming but also a difficult and challenging experiment. In contrast, the inward-open model structure of AtDTX14 is based on the inward-open crystal structure of the known homologous transporter, which is sufficient to support the main topic in our manuscript..."

I disagree with the author's opinion that the generation model of based on a known homologous transporter (MurJ) is both necessary and sufficient to support the main topic of the manuscript.

For the Cys-crosslink analysis to support our inward-facing model and the Cys modification analysis using MTSES to probe the solvent accessibility, we have to create a Cys-less mutant by changing the native Cys to other amino-acid residues. In order to respond the Referee's comment, we tried to create the Cys-less mutant of AtDTX14, by changing the 9 Cys residues within the wild-type protein to Ala, Val, and Ser. The results showed that the introduction of one Cys-to-Ala/Val/Ser mutation decreased the expression level to ~70% or less of that of the wild-type protein, and finally we observed almost no expression in the mutant with 8 Cys substitutions (C73A C92A C119A C233S C263S C292A C441A C188A).

We also tried other combinations of mutations, but the situation was similar. These results strongly indicate that the preparation of the Cys-less AtDTX14 protein is almost impossible. We also tried the cross-linking analysis of the extracellular gate using the wild-type protein (*i.e.*, containing the native 9 Cys residues), but we could not detect the cross-linked bands in the SDS-PAGE gels, because of the aggregation due to the non-specific inter-molecule S-S bond formation through the native Cys residues. Thus, the Cys-crosslink analysis and the Cys modification analysis requested by the Referee are almost impossible.

Moreover, the actual degree of homology between the transporters is never stated. Also, the structural divergence between the two proteins (MurJ has 14 TMHs whereas both hMATE and AtDTX14 have 12) raises questions regarding the accuracy throughout the entire structure (homology models often contain regions that have higher degrees of uncertainty than others).

According to the Referee's suggestion, we included the quantitative measurements of the homology between AtDTX14 and MurJ (at both the amino-acid and tertiary-structure levels) in the legend of Fig. 3 (p.31, 1.6–8).

As for the accuracy of our model structures, the MurJ paper (Kuk et al., *NSMB* 2016) also described the outward-open model structure of MurJ, which was created based on the PfMATE structure in the outward-open state, by a similar method as our inward-open model. Given that the sequence similarity between MurJ and PfMATE resembles that between MurJ and AtDTX14, the accuracy of the outward-open model of MurJ is similar to that of our inward-open model. Moreover,

the outward-open model of MurJ was created ignoring TM13 and 14, which are absent in the PfMATE structure. As described in the response in the previous round of the revision, TM13 and TM14 of MurJ have no relation to the common function of MOP superfamily transporters, and the difference between 12TM vs. 14TM is a totally unrelated point in creating the inward-facing model of eMATE. We believe that our inward-open model of AtDTX14 has sufficient accuracy to provide important insights into the transport mechanism of eMATE, as in the case of the outward-open model of MurJ in this NSMB paper.

Also, the EC coupling values do appear to be consistent with the location of the residues defining the inward and outward gates; however, they do not validate the remaining regions of the modeled protein, which also appear to be important to their proposed mechanism.

We can predict the overall fold of AtDTX14 from only the EC pairs calculated by EVfold_membrane, without using any prior knowledge about the known structures of the MATE proteins (Fig below). This clearly indicates that the EC coupling values are also consistent with the other contacts in the AtDTX14 structure.

Fig: The model structure of AtDTX14 created by EVfold_membrane

To the reviewer's comment, "The authors do not describe how the (homology-??) model was generated", the authors gave the following hardly acceptable response,

We included the description about how we created the inward-open model structure in the revised manuscript (Line 166-167).

“To investigate the inward-open conformation of the eukaryotic MATE, we generated the inward-open model structure of AtDTX14, by separately superimposing the N- and C-lobes of AtDTX14 onto the MurJ structure (Fig. 3b, e, f).”

We apologize for our poor description of the creation of the inward-facing model structure. We revised our description about the model construction in more detail (p.31, 1.3–8; p.38, 1.18–p.39, 1.3). Given that a similar method was used for the outward-open modeling of the MurJ structure (Kuk et al., *NSMB* 2016), we believe that our method is as acceptable as that used for constructing models of transporters in the opposite state.

Without the aid of structural data (that of a ligand-bound AtDTX14), the authors appear to sometimes over-interpret mutational results

For example, “In contrast, the mutation of Gln449 (Gln443) affected both the TEA and cimetidine transport activities, and the mutation of Asn412 (Asn406) strongly reduced the TEA transport activity (Fig. 5b, c). These results suggested that Asn412 is directly involved in the recognition of TEA and TEA-like substrates (Fig. 5b, c).”

The observed effects could result from both indirect or direct effects wherein specified residues play functional or structural roles. The authors include gel-filtration data to support structural effects. However, whereas the inclusion of such show that the describe variants do indeed fold, it does not distinguish between the functional or structural effects.

To more clearly distinguish the effect of the mutations, we have to determine the crystal structures of the mutants. However, the preparation of AtDTX14 (and its mutants) requires much more time and resources, as compared to bacterial proteins. The crystallization and data collection are also very difficult. It is impossible to obtain a large single crystal of AtDTX14 to collect a complete data set from a single crystal. So, we have to collect more than 100 sub data sets from hundreds of microcrystals to obtain a complete integrated data set, which usually requires about 2-3 months. Therefore, the structure determination of the mutants is not a realistic solution, to simply verify the effect of the mutation on the structure. In contrast, the gel-filtration analysis is a *de facto* standard method to verify the structural effects of the mutants, and thus has been used in many publications; e.g., Hu et al., *Nature* **478**, 408-411 (2011). We believe that the gel filtration analysis is sufficient to support our proposal that the mutations do not largely affect the native structure of the protein.

For the proposed mechanism, the authors state that “...the protonation of acidic residues causes an electrostatic attraction between them...” This appears to be a critical feature of the proposed mechanism. This statement does not describe a likely scenario, making the reviewer skeptical of the proposed transport mechanism.

We cannot understand why the Referee felt that this statement does not describe a likely scenario, since the reason is not stated in the comment. The Referee may misunderstand that the protonation of the acidic residues introduces positive charges to the sites, which might still evoke an electrostatic repulsive force between them. To avoid such a misunderstanding, we revised our description about the structural transition model (p.14, 1.8–10).

The authors state that hydrogen-bonding distances were added to Figure 4. THEY WERE NOT!

We are very sorry for our mistake in the labels in the figure. According to the Referee's comment, we included the hydrogen bonding distances in Fig. 4.

In response to the reviewer's comment, “The authors claimed to have identified the cation binding pocket in AtDTX based on the (1) presence of two acidic residues (called a conserved acidic cluster) and (2) binding of Rb⁺ and Cs⁺ ions to NorM in a cavity lined by two homologous acidic residues. However, ligand-bound structures of NorM do not show lipophilic cations bound to the same region as the Rb⁺ and Cs⁺ ions (pdb ids: 4huk (P4P), 4hum (ET), 4hun (RHQ), 4hul (Cs), 3mku (Rb) (NorM) as well as 4lz6, 4lz9 (RHQ), 3vvp (Br-NRF) (DinF),“ the author's simply state that the previously ligand-bound structures incorrectly identify the substrate-binding sites.

Without a ligand-bound structure of their own or without concrete proof of that statement, such a case cannot be made. Moreover, the authors provide no compelling evidence (modeling or otherwise) that ligands indeed bind to their proposed site.

As the Referee's suggested, the complex structure of eMATE with its substrate is important for the demonstration of our hypothesis. However, it is very difficult to determine the complex structure of AtDTX14. As described in the above response, it requires about 2-3 months to obtain a single complete data set, since we have to collect and merge more than 100 sub data sets from microcrystals. Thus, it takes about 2-3 months just to check whether one crystallization condition actually contains the substrate-bound AtDTX14. Indeed, we attempted the complex structure determination for more than one year, using several substrates, including cimetidine, pyrimethamine, rhodamine 6G, verapamil, tetraethylammonium, resveratrol, curcumine, and flavone. However, we

never saw any clear density for the substrates until now. The complex structure determination of eMATE is excessive as an additional experiment in the revision process and beyond the scope of our manuscript. In contrast, it is scientifically beneficial to discuss a possible hypothesis of the transport mechanism, combining the present and previous results of the structural and biochemical analyses.

To the reviewer's comment, "The AtDTX14 structure was solved by molecular replacement. How did the authors obtain an unbiased electron map (shown in Supplementary Figure 3.) This should be clarified," the authors responded,

"The present crystal structure is free from the bias of the model structure (NorM-VC) used in the molecular replacement, as indicated by the statistics of the structure refinement (e.g., Rwork/Rfree), the resolution, and the quality of the electron density map. The word, "unbiased", in the Supplementary Figure 2 legend means that the bias originating from the structure factor (Fc) of the present AtDTX14 model (and not from the model used in the molecular replacement) is alleviated by introducing the model-incompleteness coefficient D and the figure-of-merit m, as compared to the legacy 2Fo-Fc map (see Bernhard Rupp, BIOMOLECULAR CRYSTALLOGRAPHY, p.618-619)."

The map that is shown by the authors is not completely free from bias and is commonly called a "cross-validated sigma-A weighted map." Maps generated with the coefficients 2mFo-DFc are not "unbiased". Composite omit maps, simulated annealing omit maps and maps generated via de novo phasing are unbiased. Maps from which the phases are generated via the phases from another structure (via molecular replacement) are biased. Structure refinement (structure modeling) itself introduces bias. Structure refinement statistics do not necessarily indicate bias, they can often reveal problems due to over-fitting, which includes many types of errors. As such, the map shown should not be called an unbiased map. The reviewer was not stating that the map was inaccurate or invalid, only that the word unbiased should not be used.

As requested by the Referee, we removed the word "unbiased" and added the composite omit map in Supplementary Fig. 2b.

Overall, the paper presents a structure of an eMATE transporter that sequence divergences from of known MATE transporters. The sequence divergences result in novel features that may or may not play direct roles in transport. Whereas the reviewer believes that authors validate the observed features, their actual functional roles are not revealed clearly elucidated by the limited amount of mutagenesis data presented. Without additional structures, including one presenting the

ligand-bound form of AtDTX14, their proposed mechanism is not compelling. This is especially true considering their description of the protonation-driven helix conformational changes.

As described in the above response, the structure determination of the complex structure of AtDTX14 is very difficult, and is beyond the scope of our manuscript. In contrast, it is scientifically beneficial to discuss a possible hypothesis of the transport mechanism, combining the present and previous results of the structural and biochemical analyses.

Furthermore, the authors might consider a mechanism wherein proton-coupled export proceeds through a state in which both protons and drugs are bound. Allosteric coupling usually requires that the binding of protons (downhill) effects the binding of drugs (uphill).

In the case of the antiporters, both the substrate and co-substrate (*i.e.*, proton) are exclusively bound to the transporter, and ensure the strict 1:1 coupling of the transport. In our mechanism, the proton and the drug substrate are also exclusively bound to the transporter. There is no state in which both the protons and drugs are bound at the same time, as shown in Fig. 8. We believe that our model is the most feasible one, given the results of the structural and functional analyses in the present and previous studies.

The reviewer is aware that mechanisms similar to the one proposed here have been previously published. However, the publication of a mechanism does not establish its validity. Thermodynamic intuition (at least, reviewers) suggest that the proposed mechanism cannot be correct (especially in light of the authors' description of the proton-related effects on protein structure).

Since the Referee did not describe the likely scenario and his/her thermodynamic intuition, it is very difficult for us to address these concerns. One possible misunderstanding by the Referee is about the order of the binding/release of the substrate and proton. The Referee might mistakenly think that the proton is first bound to the MATE in complex with the substrate, and then the bending of TM7 pushes out the substrate to the extracellular side. However, we rather hypothesize that the substrate spontaneously diffuses into and rebinds from the extracellular side (Fig. 8a,b). In this context, the proton binding may play a role to prevent the rebinding of the substrate to the transporter, which will ensure the unidirectional (uphill) movement of the substrate (Fig. 8b). The hydrogen bonding between the conserved acidic residues may lock the transporter's substrate binding site in the collapsed state to prevent this rebinding of the substrate, rather than pushing out the substrate by the attractive force between them. To clarify this point, we revised Fig. 8 and the

discussion of our manuscript (p.14). We hope that the proposed mechanism in the revised manuscript satisfies the thermodynamic intuition of the Referee.

REVIEWERS' COMMENTS:

Reviewer #1 (Remarks to the Author):

As stated in my previous two comments on this manuscript, I feel that solving this structure was a major achievement.

The first revision included additional functional experiments and clarified my major points raised.

The second revision now also includes a phylogenetic tree, which further supports the alignments and the notion that eMATE family is different from its bacterial counterparts.

With regard to the modelling of inward-facing AtDTX14, this is now clearly described, as asked for by both reviewers.

With regard to the validation of the inward-facing model, the authors convincingly demonstrate that it is impossible to generate cys-less AtDTX14, which is a prerequisite to validate the extracellular and intracellular gate by disulphide cross-linking.

Regarding substrate binding, the authors tried hard to obtain a substrate bound co-crystal structure, but failed. This finding is in agreement with their transport model, as the outward-open bent conformation (i.e. the one they crystallized) does not bind substrate.

Regarding SEC to show whether mutations have a negative impact on the structure, I agree with the authors that this kind of analysis is sound. Solving crystal structures of mutants to prove structural integrity would be an overkill.

Finally, I find the model of transport being plausible. It is very hard to fully prove a transport mechanism. Also in this case there are many assumptions made based on modelling the inward-open structure and based on substrate binding data of bacterial MATEs. Nevertheless, authors have to finally come up with a model of transport that is in line with the data they present and the current knowledge of the field, even if there is the risk that a few years later the model has to be revised as a result of novel structures or functional experiments.

In summary, I consider this work as an excellent paper for Nature communications and I am confident that it will be recognized by the MATE field as an important contribution.